# Mechanical architecture and folding of *E. coli* type 1 pilus domains

Alvaro Alonso-Caballero[1], Jörg Schönfelder[1], Simon Poly[1,7], Fabiano Corsetti [1,2], David De Sancho[3,4], Emilio Artacho [1,5,6] & Raul Perez-Jimenez [1,5]

Uropathogenic *Escherichia coli* attach to tissues using pili type 1. Each pilus is composed by thousands of coiled FimA domains followed by the domains of the tip fibrillum, FimF-FimG-FimH. The domains are linked by non-covalent β-strands that must resist mechanical forces during attachment. Here, we use single-molecule force spectroscopy to measure the mechanical contribution of each domain to the stability of the pilus and monitor the oxidative folding mechanism of a single Fim domain assisted by periplasmic FimC and the oxidoreductase DsbA. We demonstrate that pilus domains bear high mechanical stability following a hierarchy by which domains close to the tip are weaker than those close to or at the pilus rod. During folding, this remarkable stability is achieved by the intervention of DsbA that not only forms strategic disulfide bonds but also serves as a chaperone assisting the folding of the domains.

[1] CIC nanoGUNE, San Sebastian 20018, Spain. [2] Departments of Materials and Physics, The Thomas Young Centre for Theory and Simulation of Materials, Imperial College, London SW7 2AZ, UK. [3] Donostia International Physics Center, San Sebastian 20018, Spain. [4] Kimika Fakultatea, Euskal Herriko Unibertsitatea (UPV/EHU), San Sebastian 20080, Spain. [5] IKERBASQUE, Basque Foundation for Science, Bilbao 48013, Spain. [6] Theory of Condensed Matter, Cavendish Laboratory, University of Cambridge, Cambridge CB3 0HE, UK. [7] Present address: Interfaculty Institute of Biochemistry, University of Tübingen, Tübingen 72076, Germany. These authors contributed equally: Alvaro Alonso-Caballero, Jörg Schönfelder. Correspondence and requests for materials should be addressed to R.P.-J. (email: r.perezjimenez@nanogune.eu)

Bacteria initiate infection by mechanical anchoring to tissues. In the case of uropathogenic *Escherichia coli* (UPEC), one of the most common and recurrent infections[1], bacteria use long appendages called pili type 1[2,3] to attach to cells of the bladder epithelium. For successful attachment, the mechanical integrity of the pilus is crucial. The pilus is composed of four different subunit types, FimA-FimF-FimG-FimH, and it is believed that all subunits and particularly their intermolecular interactions, play an important role in the attachment of the bacterium by providing mechanical resistance to the whole pilus. However, we still do not have a complete quantitative understanding of the intermolecular interactions responsible for the mechanical design of the pilus.

Pilus domains have immunoglubulin-like structure, typical of proteins with high mechanical stability[4]. Thousands of FimA subunits form the pilus rod, a helical structure with spring-like properties[5] connected to the tip fibrillum composed by FimF-FimG-FimH (Fig. 1a). The attachment to the epithelium tissue occurs by specific binding called catch-bond of FimH to D-mannose[6]. This attachment initiates internalization and biofilm formation[1,7,8]. Pilus subunits are all linked one to another by a mechanism called β-strand complementation. This mechanism consists of a long β-strand extending from one domain to the preceding one, establishing a hydrophobic interaction that secures the chain and complements the fold of each domain for full structural stabilization[9]. In addition, all the domains contain strategic disulfide bonds, which act as mechanical locks (Fig. 1b, c). Both, catch-bonds and FimA bundle uncoiling are

reversible interactions; however, the β-strand complementation is an irreversible connection. If a single β-strand breaks in the entire chain, the pilus is irreparably lost for the bacterium, which places the β-strand complementation as the most critical interaction for the pilus. Nevertheless, a detailed description of the pilus mechanical architecture and subunits interconnection has not been reported yet.

The β-strand complementation mechanism is the result of a highly coordinated process in which periplasmic FimC acts as a β-strand donor directing the complex FimC-subunit to the FimD usher located at the bacterium outer membrane[10]. There, the polymerization of subunits by β-strand exchange and final pilus secretion occurs (Fig. 1a). The process of subunit polymerization has been described in detail with crystallographic and computational studies[11–18]. Prior to the pilus assembly, the subunits undergo an oxidative folding process catalyzed by the oxidoreductase DsbA[19]. This enzyme encounters the subunits in the periplasm as they are secreted in an extended state by the SecYEG pathway[20]. The oxidation of the subunits requires DsbA as well as the electron donor DsbB, located at the inner membrane[21]. Oxidized subunits are then delivered for FimC recruitment to the FimD usher through a mechanism called donor-strand exchange[17]. A kinetic model for the folding of pilus subunits has been proposed based on bulk assays[19], but an observation of the folding process in real time has not yet been achieved experimentally. There is a lack of detailed information regarding the stability of the Fim domains and how they achieve such stability.

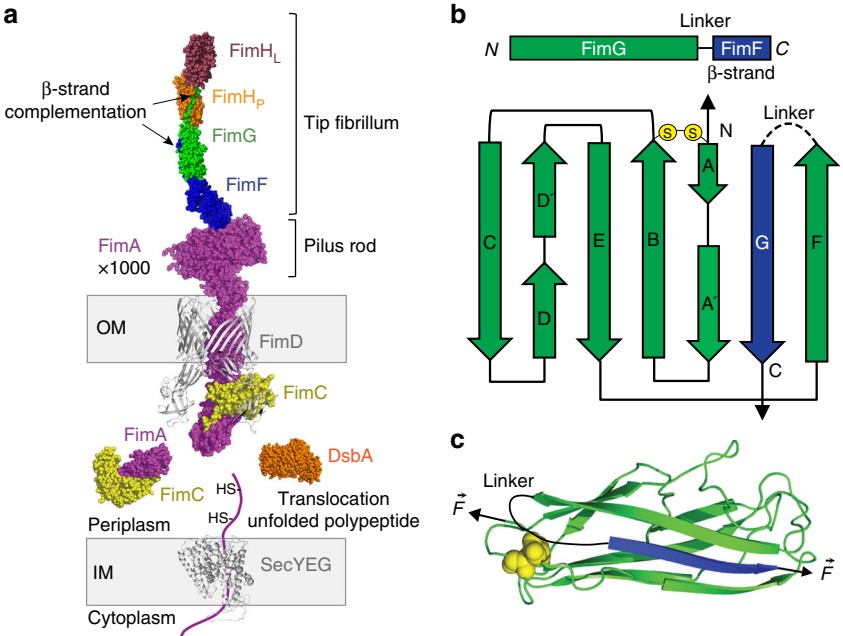

**Fig. 1** Fim domains maturation in the periplasm and incorporation into the nascent pilus. **a** Schematic view of the process of maturation of Fim proteins and their incorporation to the pilus. In this example, a FimA subunit (magenta) is translocated to the periplasmic space in a reduced (non-disulfided) and unfolded state through the inner membrane (IM) with the help of the SecYEG system (PDB: 3DIN[55]). Once in the periplasm, DsbA oxidoreductase (PDB: 1AC1[56]) induces the formation of the disulfide bond in FimA and then FimC chaperone (yellow) recognizes and binds to FimA. FimC donates a β-strand that stabilizes and helps FimA to get its native conformation. FimC-FimA complex (PDB: 4DWH[19]) then interacts with the outer membrane (OM) protein FimD/Usher (gray, PDB: 4J3O[17]). This transmembrane protein orchestrates the interchange between the β-strand of FimC with the N-terminal donor β-strand of the next subunit in the pilus, in this case another FimA. Once the exchange is done, FimA is incorporated to the pilus and FimC is released and available for another Fim protein. **b** Schematic representation of our FimG construct, which lacks its N-terminal donor β-strand. FimG domain is self-complemented through the addition in its C-terminal part, of the sequence of the N-terminal donor β-strand of FimF, the next subunit in the pilus. Between FimG and FimF donor β-strand, we place a flexible linker (DNKQ). Below, a schematic view of the self-complemented FimG is depicted. Disulfide bond is showed as two yellow circles connecting the A and B strands of FimG. **c** Cartoon representation of the self-complemented FimG with the donor β-strand of FimF (modified from PDB: 3BFQ[57]). Disulfide bond shown in yellow, flexible linker shown in black. Arrows designate how force is applied to the domain in a single-molecule force spectroscopy experiment. This vectorial force resembles the one that these domains experience in vivo

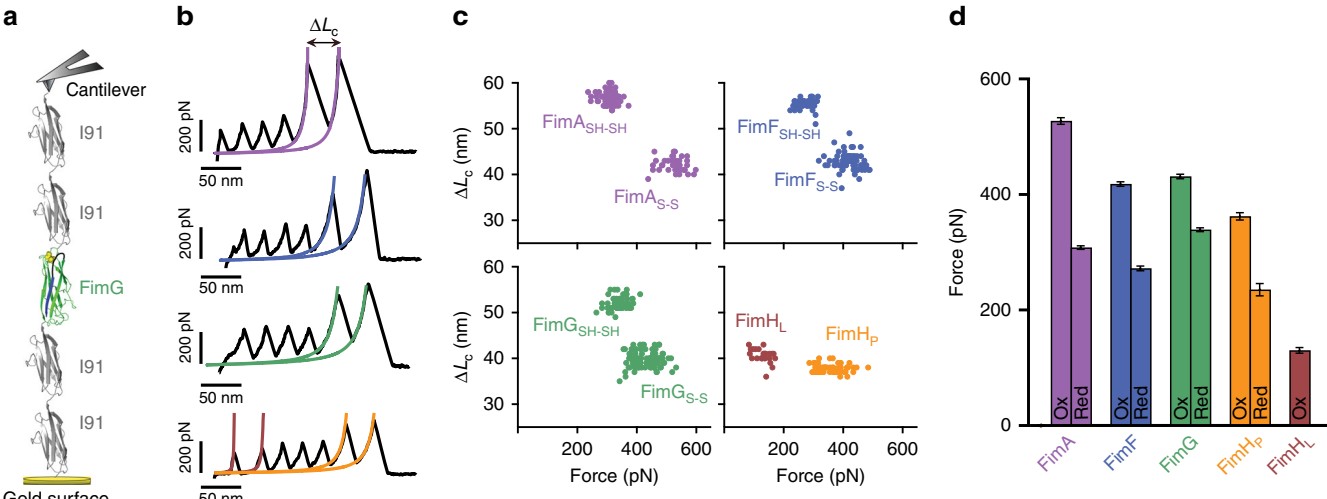

**Fig. 2** Mechanical stability of individual Fim domains. **a** Schematic view of a force-extension experiment with a polyprotein. I91 domains flank the Fim domain under study and the whole molecule is held under force between the cantilever and the gold surface. **b** Force-extension traces of the four Fim proteins. The color lines represent the fitting of the unfolding of the Fim domains with the worm-like chain model. From top to bottom are shown the traces of FimA (magenta), FimF (blue), FimG (green), and FimH, which shows seven peaks since it is composed of a pilin domain (orange) and a lectin domain (brown). **c** Contour length increment ($\Delta L_c$) vs force scatter plots of the four Fim proteins. FimA, FimF, and FimG disulfide-bonded domains (indicated with the subscript S-S) show lower $\Delta L_c$ and higher unfolding forces than when they are reduced (SH-SH subscript). FimH shows two populations that correspond with the pilin (FimH$_P$) and the lectin (FimH$_L$) domains. (FimA$_{S-S}$ $n = 43$, FimA$_{SH-SH}$ $n = 68$, FimF$_{S-S}$ $n = 79$, FimF$_{SH-SH}$ $n = 40$, FimG$_{S-S}$ $n = 110$, FimG$_{SH-SH}$ $n = 65$, FimH$_P$ $n = 51$, FimH$_L$ $n = 26$). **d** Unfolding forces of pilus domains in their oxidized (Ox) and reduced (Red) state. Histogram represents the average value of the data points shown in (**c**) and the error bars the SEM (mean ± SEM)

Here, we use a single-molecule atomic force microscope (AFM) to investigate the mechanical interactions and formation of the pilus domains. We find a hierarchical mechanical organization of the pilus, in which the resistance of the subunits to force decreases from the pilus rod to the tip. We have complemented the AFM measurements with Steered Molecular Dynamics (SMD) simulations, which give an atomic-level view of the unfolding process. This extraordinary resistance is achieved by a folding process of pilus subunits assisted by DsbA and FimC. We monitor this process on single Fim domains confirming the previously described process[8,19]. However, by the direct conversion of individual pilus subunits from the fully extended to the folded state, we discover that DsbA not only acts as an oxidoreductase but also as a chaperone, assisting pilus subunits in their folding. In summary, we provide a detailed nanomechanical description of the pilus chain in terms of mechanical design and subunit maturation. This description complements previous information to provide a qualitative and quantitative overview of how bacteria generate pili for strong attachment to tissues in order to initiate infection.

## Results

**Mechanical stability of β-strand connections in pili**. For our AFM measurements, we designed polyprotein constructs containing the pilus type 1 subunits, including the complementing β-strand from the preceding domain in the pilus chain. Given that this β-strand does not establish a covalent bond, in our constructs, we linked the β-strand to the preceding subunit through a flexible linker (Fig. 1b). This β-strand complementation, previously used for FimH[22], results in an autonomously folded domain with no disruption of the original hydrophobic interaction. In Fig. 1b, c, we show the construct for FimG complemented with the β-strand of FimF. We have performed the same self-complementation procedure for all four subunits (see Protein sequence section in Supplementary Note). Without this strategy, it would not be possible to study the pilus domains in the AFM

because the detachment of the β-strand would be undistinguishable from cantilever or surface detachment. This type of construct also ensures that the force transmission vector across the domain is the same that the pilus would experience in vivo. Therefore, by applying force to the subunit termini in the direction represented by the arrows in Fig. 1b, c, we can trigger the rupture of the interactions of the β-strand of each domain. Using the complemented Fim subunit constructs, we generated polyproteins incorporating dimeric handles of the I91 domains (formerly I27) of human titin, which serves as mechanical fingerprint due to the very well known properties determined by AFM[23] (Fig. 2a). We generated a total of five polyproteins with all pilus domains β-strand complemented: (I91)$_2$-FimX-(I91)$_2$, with X being H, G, F and A, or the array F-G-H.

In the AFM, the polyproteins were stretched in the force-extension mode at a constant speed of 400 nm s$^{-1}$. The AFM force-extension experimental data is characterized by a typical sawtooth pattern where each peak corresponds to the unfolding of one domain, except the last one, which corresponds to the detachment of the protein from the cantilever. In all the measurements, we identified four identical peaks corresponding to the unfolding of the I91 domains. The additional peak observed in each trace corresponds with the unfolding of the self-complemented Fim domains (Fig. 2b). For I91, we measured its characteristic mechanical stability of ~200 pN and contour length of 28 nm[23]. Figure 2c shows contour length increment ($\Delta L_c$) vs unfolding force plots for each pilus domain. To estimate $\Delta L_c$, we fitted the peaks using the worm-like chain model of polymer elasticity[24] (Fig. 2b).

Starting with FimA, we measured an average unfolding force of 527 ± 6 pN (mean ± SEM). We attribute this force to the rupture of the β-strand interaction, confirming the high mechanical stability of bacterial adhesion proteins[25]. This is clear by comparing the measured and theoretical contour length starting from the unraveling of the β-strand. We determined a contour length of 42 ± 2 nm (mean ± SD), which is in agreement with the theoretical length of about 44 nm estimated by considering

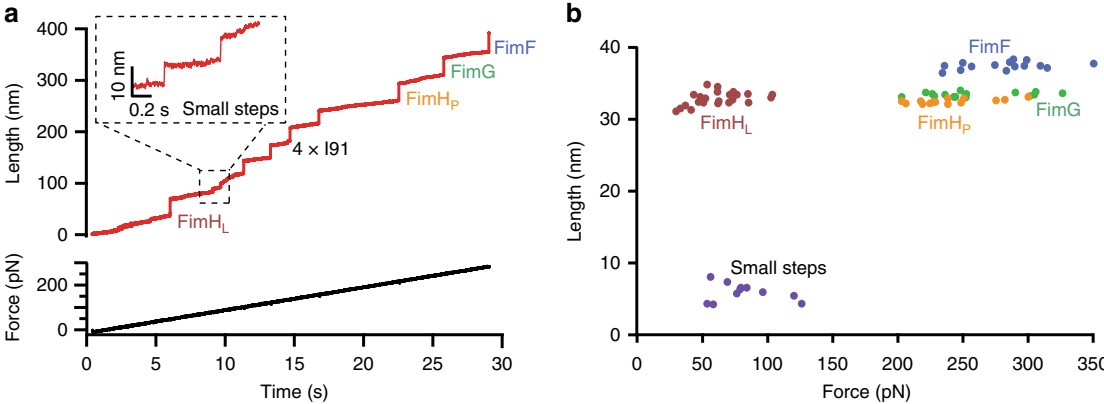

**Fig. 3** Mechanical stability of the FimF-FimG-FimH array. **a** Force-ramp experimental trace of the polyprotein (I91)$_2$-FimF-FimG-FimH-(I91)$_2$ at 10 pN s$^{-1}$ pulling speed showing the gradual unfolding of Fim domains, starting with the weaker FimH$_L$. The four I91 domains follow the unfolding of FimH$_L$ that occurs before the unfolding of FimG and FimF. Small steps of about 6 nm, as depicted in the inset, are observed in about 50% of the traces. **b** Histogram of step size vs unfolding force where the different domains can be distinguished by the color code given their step size. The average step size (mean ± SD) and unfolding forces (mean ± SEM) are: 62 ± 4 pN and 33 ± 1 nm for FimH$_L$ ($n = 23$); 238 ± 7 pN and 32 ± 1 nm for FimH$_P$ ($n = 15$); 261 ± 9 pN and 33 ± 1 nm for FimG ($n = 16$) and 281 ± 8 pN and 37 ± 1 nm for FimF ($n = 15$). The small steps show unfolding force of 82 ± 7 pN and length increments of 6 ± 1 nm ($n = 11$)

0.4 nm per residue and 5 nm for folded domain[26]. The measured length corresponds to the length of the protein in the oxidized state, i.e., with the disulfide bond formed being unbreakable at these forces. Next in the chain is FimF for which we estimated an unfolding force value of 418 ± 4 pN and a contour length of 43 ± 2 nm, again corresponding to the length of the oxidized domain. For FimG, we obtain 431 ± 4 pN for unfolding force and 40 ± 2 nm for contour length, consistent with our previous report[27]. Finally for FimH, we determined two values, corresponding to the two subdomains that compose FimH (Fig. 1a), which are the pilin subdomain (FimH$_P$), and the lectin subdomain (FimH$_L$). For FimH$_P$ the rupture force is 362 ± 6 pN and 38 ± 1 nm of contour length, and for the FimH$_L$, we measured an unfolding force of 130 ± 5 pN and a contour length of 40 ± 2 nm. Interestingly, we collected traces displaying an intermediate. Now FimH$_L$ is splitted into a small peak of 6 ± 1 nm and 98 ± 8 pN, preceding a bigger one of 36 ± 1 nm and 107 ± 5 pN of force, which are treated individually (Supplementary Fig. 1).

From the data, it is clear that FimA has the highest stability followed by FimG and FimF, with a similar force, and then by the lowest value of FimH$_P$. The final value for FimH$_L$ subdomain does not correspond to β-strand complementation rupture since this domain is covalently linked to FimH$_P$. These values establish a hierarchical mechanical design, making the pilus domains mechanically weaker as they are closer to the tip (Fig. 2d). The four Fim proteins contain a conserved disulfide bond that staples the A and β-strand in all the domains. FimH$_L$ also possesses a disulfide bond connecting strands 1 and 4b[28]. As we previously reported for FimG[27], the stretching of non-disulfide domains should yield unfolding peaks with higher contour lengths and lower mechanical stability. We measured the mechanical stability of FimA, FimF, and FimH in their reduced state and we confirmed that the non-disulfide domains are less stable than the oxidized ones (Fig. 2d). Supplementary Table 1 summarizes the unfolding forces and contour lengths of the Fim proteins in their oxidized and reduced state.

In the case of the array FimF-FimG-FimH, they do not seem to establish interactions beyond the β-strand complementation. However, other than the crystal structure, there is no experimental evidence that rules out the possibility of quaternary contacts. In order to investigate such possibility, we constructed a polyprotein with chained domains (I91)$_2$-FimF-FimG-FimH-

(I91)$_2$ to be tested in the AFM. Each domain is in the oxidized state and was β-strand-complemented as we did with the individual domains. The limitation of this particular polyprotein construct is that the unfolding of multiple domains may compete with detachment from cantilever and/or surface, due to the high mechanical resistance of each domain. To minimize such possibility, we performed the experiments in the force-ramp mode. In this mode, the force is linearly increased from −10 to 400 pN at a constant loading rate of 10 pN s$^{-1}$, in a totally controlled way by the feedback loop. The unfolding forces of all domains are lower and the full unfolding can be captured more easily[29].

In Fig. 3, we can see that the polyprotein unfolds progressively. Each domain unfolding is monitored as a step (Fig. 3a). As expected, the first unfolding event corresponds to FimH$_L$ with step size of 33 ± 1 nm and unfolding force of 62 ± 4 pN. Lower forces than those in force extension mode are expected given the lower loading rate. Following this, we detect the four equal I91 domains. The more stable domains FimH$_P$, FimF, and FimG unfold later at forces beyond 200 pN. FimH$_P$ unfold before which can be distinguished by the step size of 32 ± 1 nm and unfolding force of 238 ± 7 pN. Following this, we have the unfolding of FimG and FimF. These domains can also be distinguished by the step size; 37 ± 1 nm for FimF and 33 ± 1 nm for FimG (Fig. 3b). In force-extension experiments, the unfolding force between these domains is quite similar (Fig. 2d), however in the force-ramp experiments, FimF appears after FimG in most traces. The average unfolding forces are 281 ± 8 pN for FimF and 261 ± 9 pN for FimG. We speculate that this might be due to an interdomain interaction that is only detectable in the array of domains. Thus, force-ramp allows us to better define the mechanical hierarchy of the pilus as FimA > FimF > FimG > FimH$_P$ > FimH$_L$, which actually follows the domain sequence.

Surprisingly, we observed that about 50% of the traces show repetitive small steps of 6 ± 1 nm with an average unfolding force of 82 ± 7 pN. These steps do not correspond with the intermediate observed in FimH$_L$ for two reasons, first we often observe two steps of 6 nm but there is only one FimH$_L$; second, the step size of FimH$_L$ corresponds to the full unfolding of the domain at that force. These steps seem to be the result of a relatively weak quaternary interactions between domains. We hypothesize that these steps are essentially similar to those found in the uncoiling

of FimA of about 5 nm[30], suggesting that a similar interaction may be possible in the tip fibrillum, although the β-strand interconnection still dominates the overall mechanical stability of the fibrillum.

To find the molecular origin of this hierarchical organization of the pilus domains, we have complemented our experiments with SMD simulations. We have replicated the experimental design, producing self-complemented three-dimensional (3D) structures for the four Fim domains, in each case covalently linking the domain to the donated β-strand of the preceding one with the addition of the same linker as used in experiments. The proteins were placed in a simulation box with explicit TIP3P water and salt ions. After equilibration, the ends of the protein were stretched at a rate of $1\,\mathrm{m\,s^{-1}}$ for 10 ns. The resulting force-extension curves are shown in Supplementary Fig. 2. The simulations show a qualitatively similar unfolding mechanism for all four domains: the stretching results in two distinct ruptures in quick succession, firstly between the donated β-strand and the parallel strand of the main domain on one side of it (G and A, respectively, in Fig. 1b), and then with the antiparallel strand on the other side (G and F). This results in a double peak in the force-extension curve, which is not resolved by experiment. The first of the two peaks is consistently the highest. After the complete rupture of the β-strand complementation, the force decreases to a low value but not to zero due to the presence of the linker; the continued stretching would therefore result in the unraveling of the entire domain up to the covalent disulfide bond. We also carried out an identical simulation for FimH without the linker in order to examine its effect (Supplementary Fig. 3). In this case, the force returns to zero after the double peak as the donated β-strand is completely detached from the domain.

However, we see no substantial difference in either the unfolding dynamics or the mechanical stability of the β-strand complementation, suggesting that the effect of the linker on our analysis of the properties of the system is negligible.

**Folding of FimG assisted by FimC.** The translocation of each pilus subunit from the cytoplasm to the periplasm occurs through the membrane SecYEG pathway[9]. The SecYEG pore is about 6 nm long and <2 nm in width. This implies that each preprotein has to traverse the pore in an extended state before reaching the periplasm, which requires the preproteins to be reduced[31]. The proteins reach the periplasm where interaction with DsbA and FimC occurs (Fig. 1a). Although it is possible that the preprotein start refolding as soon as it leaves the pore, the AFM experiment is perhaps the closest rendition to the scenario of the extended protein traversing the pore. This scenario places the AFM as a suitable tool to investigate the folding process of pilus subunits assisted by DsbA and FimC for the formation of their relevant mechanical elements, disulfide bond, and β-strand. With this in mind, we first investigated how FimC assists the folding of FimG through β-strand donor exchange mechanism. We use FimG as a model, although it is reasonable to assume that the mechanism is the same for all domains.

We used the AFM in force-clamp mode in which the applied force on the protein is held constant using an electronic feedback loop, which allows to monitor the extension, collapse, and folding of an individual protein[32]. We used the polyprotein $(I91)_2$-FimG-$(I91)_2$ to monitor the complete unfolding and refolding process of the FimG subunit using its oxidized and reduced form in the absence and in the presence of the FimC protein (Fig. 4a).

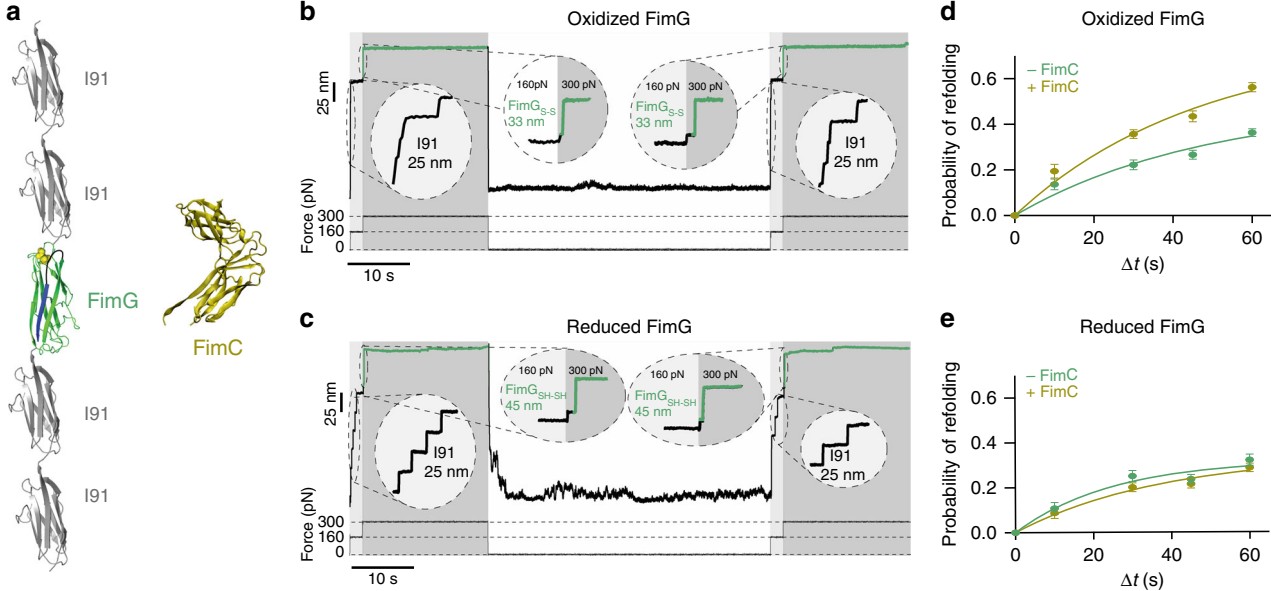

**Fig. 4** Refolding of oxidized and reduced FimG in the presence of FimC. **a** FimG construct and FimC chaperone. **b** Force-clamp trace of a disulfide-bonded FimG (FimG$_{S-S}$). Protein was submitted to a 3-pulse protocol: 160 pN for 2 s + 300 pN for 20 s, and 0 pN to allow the refolding of the protein (in this case for 45 s). To test if FimG refolded, the first two pulses were applied again. I91 unfolding steps appear as a 25 nm increase in length, meanwhile oxidized FimG yields a 33 nm step size increase. **c** Force-clamp trace of a reduced FimG (FimG$_{SH-SH}$), under the same conditions as in (**b**). The non-disulfided FimG produces a 45 nm increment in length. **d** Refolding probability of oxidized FimG at different quenching times in the absence (green, for $\Delta t = 10$ s, $n = 22$; $\Delta t = 30$ s, $n = 18$; $\Delta t = 45$ s, $n = 15$; and $\Delta t = 60$ s, $n = 11$) and in the presence of FimC (yellow, for $\Delta t = 10$ s, $n = 36$; $\Delta t = 30$ s, $n = 14$; $\Delta t = 45$ s, $n = 23$; and $\Delta t = 60$ s, $n = 16$). Exponential fittings revealed refolding rates of $0.021 \pm 0.008\,\mathrm{s^{-1}}$ (−FimC) and $0.021 \pm 0.005\,\mathrm{s^{-1}}$ (+FimC). **e** Refolding probability of reduced FimG at different quenching times in the absence (green, for $\Delta t = 10$ s, $n = 28$; $\Delta t = 30$ s, $n = 24$; $\Delta t = 45$ s, $n = 17$; and $\Delta t = 60$ s, $n = 25$) and in the presence of FimC (yellow, for $\Delta t = 10$ s, $n = 23$; $\Delta t = 30$ s, $n = 15$; $\Delta t = 45$ s, $n = 14$; and $\Delta t = 60$ s, $n = 14$). The exponential fittings revealed a refolding rate of $0.04 \pm 0.014\,\mathrm{s^{-1}}$ (−FimC) and $0.027 \pm 0.011\,\mathrm{s^{-1}}$ (+FimC). Bars show the ratio between the trajectories showing refolding and the total number of trajectories. Error bars show the SD of a binomial distribution

Therefore, we first applied a force of 160 pN for 2 s to trigger the mechanical unfolding of the I91 domains, which serve as a molecular fingerprint, each one yielding an individual step length of ~25 nm. Then, we increased the force to 300 pN for 20 s to unfold the FimG domain. We monitor unfolding step of ~33 nm for the oxidized FimG (Fig. 4b) and ~45 nm for the reduced form (Fig. 4c). Once FimG was stretched during the first force probe pulse, the force was quenched to 0 pN, which allows FimG to refold in the absence or in the presence of FimC. In the second force probe pulse, which was identical to the first pulse, we unfolded the I91 domains as well as the FimG domain in order to identify its proper folding during the quenching time. The absence of refolding can be also monitored after the quenching time, showing no FimG unfolding step in the second pulse (Supplementary Fig. 4).

We applied quenching times of 10, 30, 45, and 60 s (Fig. 4 and Supplementary Fig. 5) and determined the corresponding refolding probability for FimG. More than ten traces per refolding time and condition were collected. In Fig. 4d, e, the probability of refolding of oxidized FimG ($FimG_{S-S}$) and reduced FimG ($FimG_{SH-SH}$) is shown, both in the presence and in the absence of the chaperone FimC. As we increase the quenching time, the probability of refolding increases for $FimG_{S-S}$. Our FimG domain is self-complemented with the donor β-strand of FimF, therefore, it may be able to fold also in the absence of an external help. In the presence of FimC, the probability of refolding of $FimG_{S-S}$ is higher, becoming noticeable from 30 to 60 s of quenching time (Fig. 4d). This is in agreement with previous observations in bulk experiments for FimA[33]. However, this effect is not detectable in $FimG_{SH-SH}$ whose probability of refolding is barely affected in the presence of FimC (Fig. 4e). For $FimG_{S-S}$ (Fig. 4d), calculated rates do not show differences between the refolding experiments in the absence and in the presence of FimC, being in both cases around 0.02 s$^{-1}$. For $FimG_{SH-SH}$ (Fig. 4e), the rates are in the same range, being 0.03 s$^{-1}$ in the presence of FimC and 0.04 s$^{-1}$ in the absence of it. Overall, these results demonstrate that FimC seems to recognize and increase the probability of refolding of disulfide-bonded FimG domains, even if it does not accelerate the kinetic rate constant for FimG refolding. This is in agreement with previous findings in bulk[19].

We also estimated the unfolding rates of reduced and oxidized FimG at 300 pN. The graph shown in Supplementary Fig. 6 represents the normalized extension of the summed traces vs time. Both datasets were fitted to a single exponential function. The data clearly indicates that reduced FimG unfolds around 30 times faster than oxidized FimG at 300 pN ($FimG_{S-S}$: $1/\tau_1 = 0.94$ s$^{-1}$; $FimG_{SH-SH}$: $1/\tau_1 \sim 29.9$ s$^{-1}$). This finding correlates with the force-extension data obtained for reduced and oxidized FimG, where the reduced protein showed lower mechanical stability than the oxidized one.

**Oxidative folding of FimG assisted by DsbA**. The interaction with DsbA has been suggested to be the very first process occurring in the periplasm after translocation of pilus domains, as Fim proteins oxidation seems to be crucial prior to the interaction with FimC[19]. To measure the refolding of FimG in the presence of DsbA, we performed force-clamp experiments following a similar protocol than for FimC, but using DsbA to monitor oxidation of disulfide bonds (Fig. 5a). During these measurements, we use the reduced form of FimG, which is oxidized by $DsbA_{S-S}$ during the quench. The applied force protocol consisted of a 2 s pulse at 160 pN followed either by one pulse of 7 s at 300 pN or two pulse forces, one of 300 pN and 5 s followed by another of 100 pN during 2 s (Fig. 5b). The 2 s pulse at 100 pN was used in order to favor the possible reduction of $FimG_{S-S}$ by

$DsbA_{SH-SH}$, since the redox equilibria of both proteins changed along the experiment and enzymatic reduction of disulfides is impaired at high force[34].

To prevent non-oxidative release of DsbA from its substrate[35], the time at 300 pN was only 5–7 s. After the stretching time at 300 pN (or 5 s at 300 pN + 2 s at 100 pN), the force was again quenched to 0 pN for 45 s, before repeating the same force protocol. The resulting force-clamp traces indicate two unfolding patterns depending on the initial redox state of the FimG. A ~33 nm step corresponds to the unfolding length of oxidized FimG (Fig. 5b). If a reduced DsbA ($DsbA_{SH-SH}$) cleaves the disulfide bond of the partially unfolded FimG, an additional ~12 nm step appears accounting for the sequestered residues. The sum of both steps gives the total length of 45 nm. The trace in Fig. 5b shows the sequence unfolding/reduction during the first pulse, folding/oxidation during quenching and unfolding/reduction in the probe pulse. In Fig. 5c, we determined the refolding probability of FimG under different conditions. The presence of DsbA increases the refolding probability of FimG about threefold when compared with the other conditions (absence and presence of FimC). This result indicates that DsbA activity is not only limited to the formation of disulfide bonds, moreover it possesses chaperone activity on its substrate FimG. Nevertheless, this chaperone activity is also observed in the absence of oxidase activity. We collected traces in which DsbA was present but no formation of disulfide bond was detected, which means that DsbA did not show oxidase activity (Fig. 5d). Even in such case, we still observe refolding of FimG with a high probability, which contrasts with the observations in the absence of DsbA (Fig. 5e). This foldase activity of DsbA was observed for both reduced and oxidized FimG (Supplementary Fig. 7).

**Folding of individual FimG co-assisted by DsbA and FimC**. We thus investigated the unfolding and refolding behavior of FimG in the presence of both FimC and DsbA. These measurements intend to reproduce the conditions that pilus domains encounter in the periplasm. Here, the force protocols were identical to those used in the case of FimG and DsbA. Figure 5f shows a force-clamp trace in which an oxidized FimG is reduced upon interaction with DsbA. After 45 s at 0 pN, FimG appears again oxidized, indicating that during the quenching time it was reoxidized by DsbA. Figure 5c, g show the refolding probability under different conditions and the probability of successful disulfide bond formation, respectively. As expected, the presence of DsbA enables the oxidation of FimG, something only possible if an oxidase is present in the medium. However, when compared with the data collected for the reduced FimG, it is easy to see that the presence of DsbA increases over three times (around 70%) the refolding probability of FimG (around 20%), something that not even FimC does on oxidized FimG (Fig. 4d). These findings point to DsbA as the main contributor, not only to disulfide bond generation but also to folding. We believe the mechanism of action of FimC and DsbA is expected to be identical for all domains.

**Folding of $FimH_P$ vs $FimH_L$**. In the pilus, FimH is the domain that establishes actual contact with the host through the $FimH_L$ subdomain. Given that $FimH_L$ is the only domain in the chain that does not incorporate a complementing β-strand from a different domain, it is also the only one that can potentially unfold and refold in the pilus upon force application. Any other domain unfolding would result in β-strand disconnection and pilus rupture. In the outer space, unfolding of $FimH_L$ could be possible under extreme forces. This implies that $FimH_L$ will have to refold quickly in order to recover its functionality for

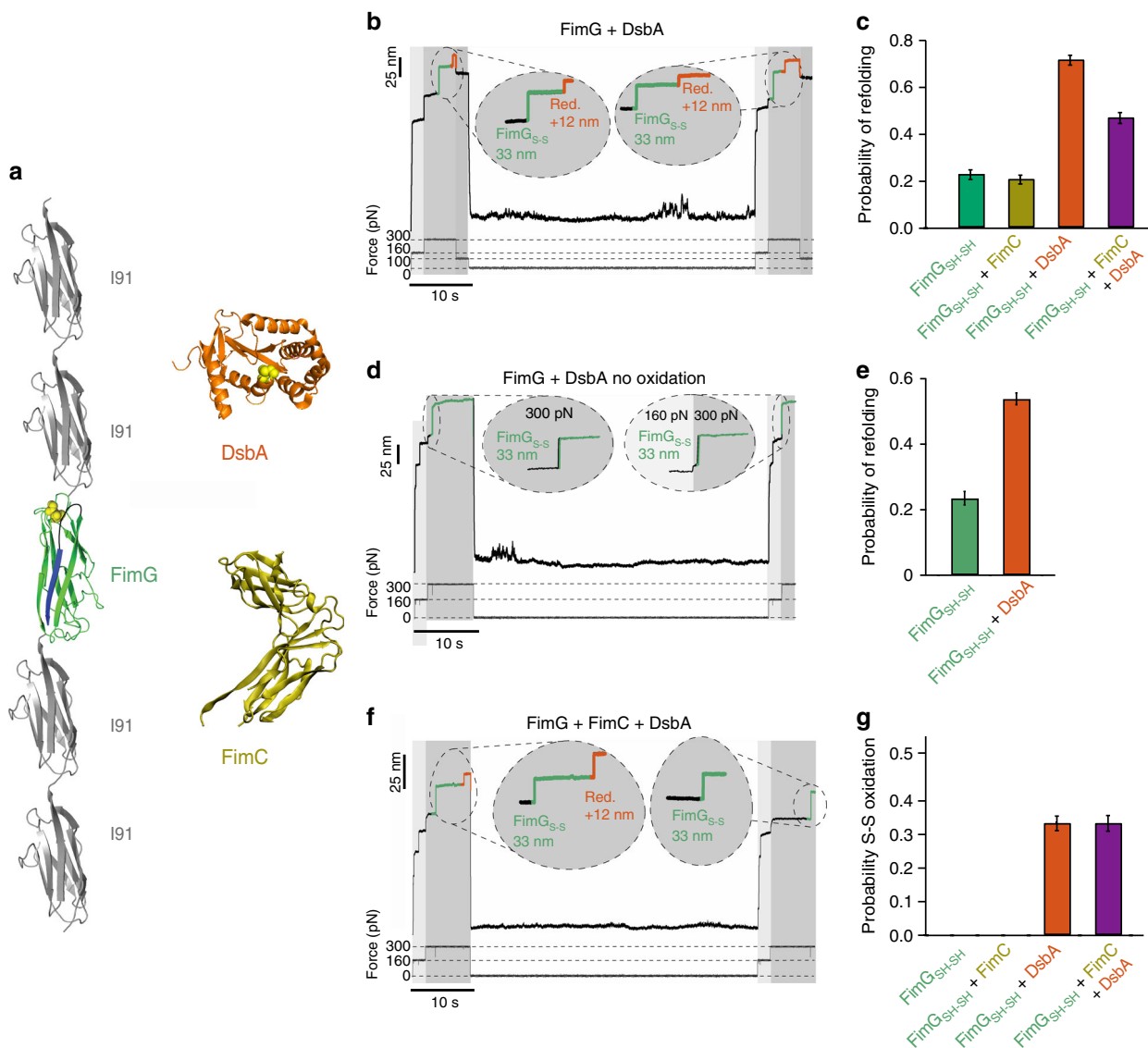

**Fig. 5** Oxidative folding of FimG in the presence of DsbA and FimC. **a** FimG construct, periplasmic FimC, and DsbA oxidoreductase. **b** Force-clamp trace of FimG in the presence of DsbA. A 4-pulse protocol was applied to the protein. First, 160 pN for 2 s (I91 fingerprint detection); second, force was increased to 300 pN for 5 s (or 7 s) for FimG unfolding. Third, force was quenched for 2 s to 100 pN in order to ease the reduction of the protein by DsbA. Fourth, force was quenched to 0 pN for 45 s for refolding. In order to test if FimG refolded and if its disulfide bond was reformed, we applied the same pulse protocol. I91 unfolding steps appear as a 25 nm increase in length, while oxidized FimG yields a 33 nm step size increase, and its reduction produces a 12 nm increment in length. **c** Refolding probability of FimG after 45 s of quenching time under different conditions (FimG$_{SH-SH}$, $n = 16$; FimG$_{SH-SH}$ + FimC, $n = 14$; FimG$_{SH-SH}$ + DsbA, $n = 16$; FimG$_{SH-SH}$ + FimC + DsbA, $n = 21$). **d** Force-clamp trace for FimG in the presence of DsbA but with no oxidase activity. In this case a 3-pulse protocol was applied. **e** Probability of refolding of FimG$_{SH-SH}$ alone or with DsbA with no oxidase activity. **f** Force-clamp trace of FimG in the presence of both FimC and DsbA. **g** Percentage of disulfide bond reformation of FimG after 45 s of quenching time under different conditions. Error bars show the SD of a binomial distribution

attachment. To test this idea, we have investigated the folding kinetics of FimH$_L$ and compared to that of FimH$_P$. Using the polyprotein (I91)$_2$-FimH-(I91)$_2$ (Fig. 6a), we have applied a four-pulse protocol with three unfolding forces, 60 pN to monitor FimH$_L$ unfolding followed by a 160 pN pulse for I91 and a 250 pN for FimH$_P$ unfolding. We then quenched to 0 pN to trigger refolding. In Fig. 6b, we observe that FimH$_L$ subdomain unfolds in the first pulse with step size of 33 nm, folowed by the four I91 unfolding in the second pulse, and finally the FimH$_P$ subdomain at 250 pN with a step size of 32 nm. This is the expected sequence according to force-extension experiments in Fig. 2. In the second force test, the FimH$_L$ subdomain is again observed at 60 pN, but the FimH$_P$ subdomain is now observed at 160 pN, indicating that

it might not have recovered full stability by itself during the quenching, which contrasts with FimH$_L$. We have estimated the refolding probability of FimH$_L$ and FimH$_P$ at 10 s quenching time. As expected, FimH$_L$ shows much higher, about twofold, folding probability indicating that it recovers quicker than FimH$_P$ (Fig. 6c).

## Discussion
Here we report a mechanical characterization of the type 1 pilus domains complemented with their cognate β-strands. This protein design resembles the interactions present in a real pilus. Using an AFM, we are able to study their mechanical integrity.

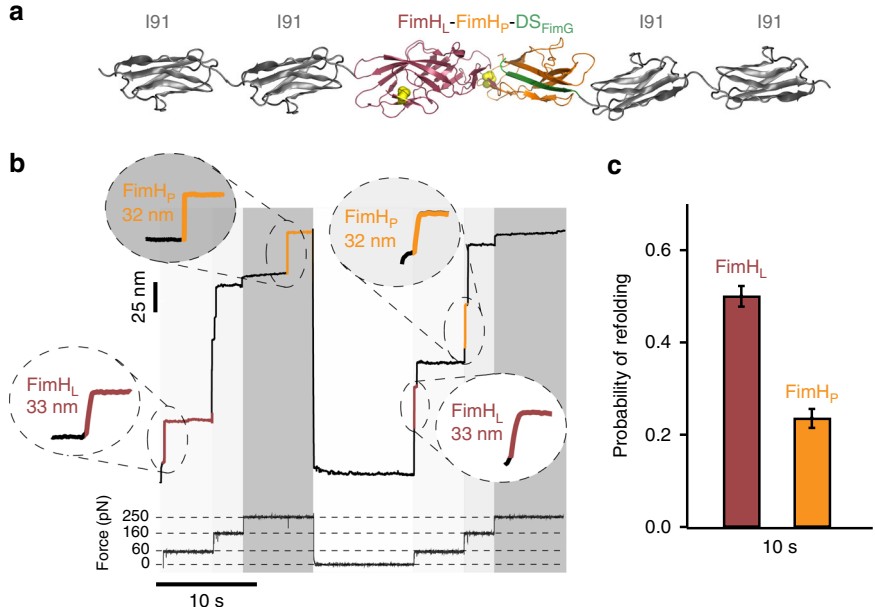

**Fig. 6** Folding of oxidized FimH$_L$ vs FimH$_P$. **a** Schematic representation of the polyprotein (I91)$_2$-FimH-(I91)$_2$ construct. **b** Force-clamp trace of FimH domain. Protein was submitted to a 4-pulse protocol: 60 pN for 5 s + 160 pN for 3 s + 250 pN for 7 s, and 0 pN for 10 s to allow the refolding of the protein. To test if FimH refolded, the first three pulses were applied again. I91 unfolding steps appear as a 25 nm increase in length, meanwhile oxidized FimH$_L$ and FimH$_P$ yield both a 33 and 32 nm step size increase, respectively, and can be readily differentiated because of the unfolding force pulse regimes. **c** Refolding probability of FimH at 10 s of quenching time (FimH$_L$, $n = 20$ and FimH$_P$, $n = 17$). Bars show the ratio between the trajectories showing refolding and the total number of trajectories. Error bars show the SD of a binomial distribution

We found that the pilus domains exhibit high mechanical stability, among the highest known for proteins[36]. Our findings also highlight the fact that pilus domains are mechanically weaker as we approach the tip fibrillum. FimA was found to be the highest mechanically stable protein in the pilus. The strongest interaction between FimA domains is perhaps mandatory since it is the most present interaction in the pilus. Pilus dissasembly due to rupture of FimA–FimA interaction could lead to cell detachment, hence it makes it plausible that the β-strand complementation strength has evolved in such a way that the most numerous interaction is the strongest. This high mechanical stability was also reported for the shaft pilins FimA and SpaA from Gram-positive bacteria[37]. Very recently, the mechanical stability of the staphylococcal adhesin SdrG was shown to reach 2 nN[25]. All these findings indicate that the main components of bacterial pili display high mechanical stability.

Bacteria exposed to shear forces need to distribute the mechanical stress along the pilus. Catch-bonds between FimH and the mannosylated glycoprotein of a bladder cell break at forces of 60–180 pN, depending on pulling conditions[38]. In order to reduce the tension on this interaction, the pilus rod unravels at forces below 100 pN acting as a shock absorber[30]. These two interactions may explain the adhesion of the bacterium, but do not seem to justify the mechanical resistence of the whole pilus. Given that the β-strand interactions are the only non-reversible interactions in the pilus, they must be higher than the catch-bond and rod uncoiling forces. The elevated mechanical resistance of the β-strands actually allows for the pilus to behave as a shock absorber because it prevents rupture and secures the spring-like function of the pilus. In addition, we have discovered mechanical signatures that suggest an additional quaternary interaction between the domains of tip fibrillum similar to that found on the FimA domains in the pilus rod. However, this interaction does not seem to impact the mechanical stability of the domains, still dominated by the β-strand complementation.

While uncoiling or detaching the pilus require low forces[30,38], all the donor β-strands and pilin domain interactions are very strong. This hierarchy of mechanical forces suggests a mechanism for pili dettachment: first, the pilus rod will completely unravel and then the catch-bond between FimH and the mannose will be broken. The latter process will lead to cell detachment but preserving the structure of the pilus, which will remain intact for a new attachment in the bladder epithelium. Considering that bacteria surface is surrounded by a large number of pili[39], this strategy seems to be the best suited for bacterial attachment. Our data also reveal that disulfide-bonded domains are mechanically more stable than the reduced ones. As we previously reported for FimG[27], the disulfide bond acts as a mechanical lock that pushes the unfolding of the domain through a higher energetic unfolding pathway. The position of this disulfide bond is conserved not only for the subunits of the type 1 pilus but also among other subunits from different pili, making it an important mechanical feature in proteins used by bacteria for attachment[40].

The SMD simulations give us insight into the events taking place during the unfolding of the domains, showing a sequential ripping of the hydrogen bonds keeping the donor β-strand in place. A map of the persistent hydrogen bonds present in each domain from simulation (Supplementary Fig. 8) shows a qualitatively very similar picture for all cases; however, we find a surprising correlation between the number of hydrogen bonds in the β-strand complementation structure and its experimentally measured mechanical stability: FimA has the most hydrogen bonds (20) and has the highest stability, while FimH has the least (17) and the lowest stability. This suggests a clear path for further investigating the contribution from individual residues.

Our folding experiments suggest that FimC is oriented to protect and stabilize the domains before their incorporation into the pilus, as it has been already pointed out[41]. This role of FimC could be crucial in a crowded and packed space as the periplasm,

to avoid aggregation or donor-strand complementation between immature and unstable subunits. We found that in the absence of disulfide bonds in FimG, the chaperone effect of FimC is null. This confirms that the oxidation of FimG is a prerequisite for FimC. Previous work highlighted the disulfide bond creation as a quality test used previous to subunit incorporation to a growing pilus[19]. This difference between disulfide-bonded and non-disulfide-bonded domains could play an important role when considering that reduced domains are mechanically weaker than the oxidized ones. Hence only disulfide-bonded mechanically stable domains stabilized by FimC would be finally incorporated into the pilus. Thus, the presence of the disulfide bond is important for the mechanical stability of the pilus and for FimC recognition during pilus biogenesis.

In the case of DsbA, in our experiments, we not only see the formation of disulfide bonds catalyzed by DsbA, we also observe that DsbA increases FimG refolding in a greater extent than FimC does. This finding is a totally new aspect describing the folding of pilus proteins. DsbA would assist Fim proteins to fold not only due to the entropic and enthalpic contributions derived from the disulfide bond formation[42], but also helping the Fim proteins to collapse and form their native contacts. When DsbA crystal structure was reported[43], it was proposed that some structural features could help DsbA to bind partially folded proteins. Later, it was proposed that DsbA has chaperone activity on the maturation of PapD chaperone (FimC equivalent) of the pilus type P[44]. Further experiments exploring the peptide-binding ability of DsbA showed that the mixed disulfide intermediate between DsbA and its substrate was stabilized by non-covalent interactions[45]. Besides, the chaperone-like activity of DsbA on proteins lacking cysteines on their sequence was demonstrated[46]. Even though the chaperone activity of DsbA was already suggested, our experiments are the first demonstration on its native substrate. One could expect that if both FimC and DsbA have chaperone activity; the combination of the two of them would have a synergistic effect on the folding of FimG. However, our experiments clearly show that the combination of both has a similar effect on the refolding of FimG than just DsbA by itself, highlighting that DsbA seems to be the principal chaperone.

Our results complement previous observations to render a complete view of the mechanical architecture and maturation of the pilus. This knowledge will help to develop new therapeutical mechano-drugs that target the strength of the donor-strand complementation. Nowadays traditional antibiotic treatments are becoming less effective due to the appearance of resistant strains. Targeting some of the main actors involved in pilus mechanochemistry could be another therapeutic approach.

## Methods

**Protein expression and purification**. The genes encoding the proteins used in this article were synthesized and codon-optimized for expression in *Escherichia coli* cells (Life Technologies). These genes were cloned in the pQE80L plasmid (Qiagen) with the exception of DsbA, which was cloned in pET11a plasmid, both kindly provided by Julio Fernandez's lab. The gene sequence of the pilus subunits was built including their cognate donor-strand sequence following their C-terminal end (FimA-DS$_{FimA}$, FimF-DS$_{FimA}$, FimG-DS$_{FimF}$, and FimH-DS$_{FimG}$) with a four aminoacid sequence (DNKQ) in between acting as a linker as it was previously done[11] (Supplementary Note). Two copies of the I91 domain (former I27) from human titin protein were placed flanking the pilus subunits as it was explained above. The last C-terminal I91 domain contains two cysteine residues in its C terminal sequence, allowing polyprotein immobilization on the gold surface.

BL21 (DE3) *E. coli* competent cells (Agilent Technologies) were transformed and grown in LB media at 37 °C up to OD$_{600}$ ~0.6. Then the protein expression was induced adding 1 mM of IPTG and the culture was incubated overnight at 37 °C. After cell pelleting through centrifugation, cell lysis was performed mechanically with a french press and the His$_6$-tagged polyproteins were purified by affinity chromatography using a Talon column (Clontech). Disulfide bond formation was triggered by oxidation with 0.1% of H$_2$O$_2$ overnight at room temperature. Reduced non-disulfide proteins were obtained incubating with 10 mM DTT overnight at

room temperature. Protein purification was followed by a size exclusion chromatography in a Superdex 200HR column (GE Healthcare). The buffer used was HEPES 10 mM pH 7.0, NaCl 150 mM, and EDTA 1 mM. Protein purity was addressed with SDS-PAGE and the proteins were aliquoted and frozen until use.

In the case of DsbA, we used a different purification protocol. After protein expression induction with 1 mM IPTG, the culture was incubated overnight at 20 °C. Instead of mechanical disruption of the bacterial cells, an osmotic shock was induced in order to perform the periplasmic extraction of DsbA[47]. Further purification steps included anion exchange chromatography in 5 mL HiTrap Q FF column (GE Healthcare) and size exclusion chromatography[35].

**Single-molecule atomic force microscopy experiments**. Atomic force microscopy (AFM) experiments were conducted in a commercial Atomic Force Spectroscope (Luigs & Neumann). The cantilevers used in these experiments were calibrated using the equipartition theorem and they had typical spring constants of 15 pN nm$^{-1}$ (Bruker MLCT) and 6 pN nm$^{-1}$ (Bruker OBL-10). The proteins under study were incubated for 10 min over custom-made gold surfaces at a final concentration of 0.1–1.0 g L$^{-1}$ and the buffer used was HEPES 10 mM pH 7.0, NaCl 150 mM, and EDTA 1 mM in all the experiments.

Force-extension experiments were conducted at 400 nm s$^{-1}$, applying first a force of ~1 nN on the cantilever. Force-extension data were analyzed using the worm-like chain for polymer lasticity[24]. The contour lentgh increments are estimated as the difference between the fittings in the asymptote, which corresponds to the fully unfolded domain. Force-ramp experiments were done at 10 pN s$^{-1}$ pulling speed for 30 s. Force-clamp experiments were conducted with the FimG construct in its reduced and oxidized form and in the presence of FimC and/or DsbA, at 10 and 100 μM, respectively. Folding experiments in the presence of FimC were conducted applying two consecutive force pulses of 160 pN for 2 s and 300 pN for 20 s, followed by a quench pulse at 0 pN of different time lengths in order to allow the refolding of the protein. Then, the same two force pulses were applied again in order to check the folding of FimG. Oxidation and folding experiments in the presence of DsbA were conducted applying either three consecutive pulses of 160 pN (2 s), 300 pN (5 s), and 100 pN (2 s), or two consecutive pulses of 160 pN (2 s) and 300 pN (7 s), then followed by a quench pulse of 0 pN. After the quench pulse of 0 pN for 45 s, the force pulses were applied again. For the combined experiments with FimC and DsbA, the same pulse protocols used for DsbA alone were applied. Experiments conducted on the FimH construct were carried out using a four-pulse protocol with three unfolding forces, 60 pN (5 s) to monitor FimH$_L$ unfolding followed by a 160 pN (3 s) pulse for I91 and a 250 pN (7 s) for FimH$_P$ unfolding. Thereafter, the force was quenched to 0 pN to trigger refolding during 10 s and then the three-force pulses were applied again. In the case of force-ramp experiments conducted on the pilus tip polyprotein (FimH-FimG-FimF), the force was linearly increased from −10 to 400 pN at a speed of 10 pN s$^{-1}$. The traces obtained in both force-extension and force-clamp modes were collected and analyzed with a custom-written code in Igor Pro 6.37 (Wavemetrics), available to the scientific community upon request. All the figures were generated using Igor Pro, Adobe Illustrator CS6 (Adobe), PyMOL (Schrödinger), and VMD[48,49].

Refolding probability was calculated as the ratio between the number of trajectories showing refolding and the total number of trajectories collected for a specific quenching time and experimental condition. The standard deviation was calculated assuming data as a binomial distribution.

**Steered molecular dynamics simulations**. All-atom simulations were carried out on the self-complemented proteins with the GROMACS[50] code (version 4.6.5). Modelization of these structures was carried out based on previously resolved 3D structures accessible on the Protein Data Bank (self-complemented FimA: 2JTY, FimC-FimF-FimG-FimH complex: 4J3O). Each subunit was separated together with the donor β-strand from the next one, and the DNKQ linker was manually inserted between the two to form a single chain, equivalent to what was used in the force spectroscopy experiments. After the insertion, the proteins were placed in a box of dimensions 7 × 7 × 28 nm. The principal axis of the proteins was aligned with the z direction. Explicit water density and 0.15 M of NaCl were added to the box. The CHARMM27-CMAP[51] force field was used for the protein and TIP3P[52] for the solvent. The systems were initially prepared by a steepest descent minimization followed by an equilibration run of 100 ps using velocity rescaling with a stochastic term[53] to ensure a physically reasonable configuration. For the case of FimF, for which no structure complemented with FimA has been reported, the donor β-strand of FimA was manually inserted together with the linker and a long equilibration run was performed to ensure the correct bonding between the subunits. For the case of FimH, only the pilin domain was used. Each protein was then pulled along the z direction of the box at 1 nm ns$^{-1}$ for 10 ns with Nosé-Hoover temperature coupling. We use a spring constant of 1000 kJ mol$^{-1}$ nm$^{-2}$. The pulling reference groups were the terminating nitrogen and carbonyl carbon atoms along the main chain from the N and C-terminal aminoacids of the donor Fim-linker-β-strand complex. During the run, both the protein and solvent were kept at 300 K; no pressure coupling was applied. The simulations used a time step of 0.001 ps, periodic boundary conditions in all directions, the smooth particle mesh Ewald[54] (SPME) method for long-range electrostatics, and a real-space cutoff for van der Waals interactions of 1.5 nm.

The pulling was performed up to the removal of the donor β-strands from the protein fold. The interaction of FimH with the donor β-strand of FimG was also tested without the linker, in order to validate our experimental approach with the covalent attachment. In this case, the reference pulling groups were the N-terminal aminoacid nitrogen atom from the FimH pilin domain and the C-terminal aminoacid carbonyl carbon from the donor β-strand of FimG. For all the proteins, the hydrogen-bonding pattern was analyzed during the course of the equilibration and pulling simulations. This was done based on the instantaneous O–H distances and accounting for regular fluctuations in time.

**Data availability**. The data that support the findings of this study are available from the corresponding author upon reasonable request.

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

## Acknowledgements

Plasmid pQE80L and pET11a-DsbA were a kind gift from Professor Julio M. Fernández (Columbia University). We also thank Professor Gabriel Waksman (University College London) for protein and DNA samples. Research has been funded by the Ministry of Economy and Competitiveness (MINECO) grants BIO2016-77390-R, BFU2015-71964 to R.P.-J.; CTQ2015-65320-R to D.D.S.; European Commission grant CIG Marie Curie Reintegration program FP7-PEOPLE-2014 to R.P.-J. A.A.-C. is funded by the predoctoral program of the Basque Government. R.P.-J. and E.A. thank CIC nanoGUNE and Iker-basque Foundation for Science for financial support. D.D.S. is supported by a Ramón y Cajal grant RYC-2016-19590 from MINECO. E.A. and F.C. thank the funding provided by the grants FIS2012-37549-C05 and FIS2015-64886-C5-1-P from MINECO, and Exp. 97/14 (Wet Nanoscopy) from the Programa Red Guipuzcoana de Ciencia, Tecnologia e Innovacion, Diputacion Foral de Gipuzkoa. We acknowledge technical support provided by IZO-SGI SGIker of UPV/EHU and European funding (ERDF and ESF) for the use of the Arina HPC cluster, and the assistance provided by Txema Mercero and Eduardo Ogando.

## Author contributions

R.P.-J. conceived the project and designed research. A.A.-C. and S.P. cloned and expressed proteins. A.A.-C., J.S., S.P., and R.P.-J. performed AFM experiments and data analysis. F.C. performed SMD simulations. F.C., E.A., D.D.S., and A.A.-C. analyzed the simulations. R.P.-J., A.A.-C., and J.S. drafted the paper and all authors contributed in revising and editing the manuscript.

## Additional information

**Competing interests:** The authors declare no competing interests.

