## [Peer Review File · Nature Communications]

Reviewers' comments:

Reviewer #1 (Remarks to the Author):

Alonso-Caballero et al. describe AFM pulling measurements on the subunits of the bacterial pilus. They investigate how the unfolding forces for the different subunits compare, finding that there is a distinct hierarchy, with the domains located closer to the tip unfolding at lower force than those farther away. Separately, they also characterize how the folding of the FimG domain is affected by the co-factors FimG and DsbA, finding that they do change the refolding (especially DsbA, which increases the amount of refolding seen significantly).

The authors are studying a very interesting system, and mechanical pulling measurements are clearly relevant to understanding how pili work. The design of the β -complemented domains is clever, and the measurements are generally well done. In principle, a study like this can be of sufficient general interest and significance to be suitable for publication in Nature Communications. However, there are a number of flaws in the manuscript that make it unsuitable, at least in its current form.

The most important concern regards the design of the study presented here. The authors have previously studied the unfolding of the FimG domain. The current manuscript extends that initial work in two directions: (i) by comparing the unfolding of FimG to that of FimH, FimF, and FimA; and (ii) by looking at the refolding of FimG under different conditions. In my view, these two components each represent interesting but incremental advances over the previous work, in two different directions. Adding them together does not amount to the "complete mechanical characterization of the pilus" that the authors promise in the abstract. For a complete characterization, one would expect to investigate not only the unfolding of each of the domains separately, but also how each domain behaves in the context of a complete pilus, in order to understand the role of the full interfaces between the domains (not simply the effects of β -complementation, which are captured here). One would also expect to see a study of the refolding of more than just the FimG domain, but also the other domains. Indeed, the refolding of the FimH domain would seem to be the most important to understand, since it is the least mechanically stable domain and hence the one that should unfold and refold most frequently when the pilus is under tension. A truly complete mechanical characterization containing such additional measurements would, in my view, be suitable for publication in Nature Communications. Otherwise, the authors might consider splitting the paper into its two parts (which are not all that closely related, as it stands) and publishing each one separately in a more specialized journal.

Some additional issues to be addressed in revisions:

1. There are numerous places where incorrect English make the manuscript more difficult to read than it should be. Just a few examples from the first page of the introduction:

- line 51: instead of "full tertiary structure" I think the authors mean "full quaternary structure," given that several different domains are involved.
- line 77: "nanomechanical architecture" is an overly vague phrase that is not meaningful. Strictly, the 'mechanical architecture' is just the structure, hence 'nanomechanical architecture' really just means the atomic-scale structure, which has indeed been solved for the pilus (it's shown in Fig 1A).
- line 79: "the mechanical resistance of the pilus" would be better expressed as "the resistance of the pilus to mechanical loads"
- line 81: "the mechanical resistance of the subunits decreases as they approach to the tip" implies that the 'mechanical resistance' changes as the subunits move in space closer to the tip, which I do not believe is what the authors mean to say. Instead, the 'mechanical resistance' of the

domains located close to the tip is lower than that of the domains located farther away.

- Multiple locations: “therein” is used in situations where it is inappropriate.

Please edit the text carefully throughout to avoid incorrect word usage, incorrect grammar, and confusing sentence construction.

2. On a related note:

- The phrase “a mechanical hierarchy decreasing from the pilus rod to the tip” in the abstract is unclear.
- Please spell out acronyms in the main text, not in the abstract.
- Page 4, line 153: “non-disulfided” is not a word.

3. The first paragraph of the introduction jumps straight into the details of the structure of the pilus. It would be more effective to start by describing the problem that is being addressed in this work, before describing the background information about the pilus. Page 2, line 43: It would be helpful to reference Fig 1A (structure of pilus) here.

4. The authors use the term “atomic force spectroscope” instead of the term that is almost universally used in the literature, atomic force microscope. There is no value to be gained by renaming a common tool with some bespoke label, only confusion. Please use the standard terminology instead.

5. The authors report mean \pm S.D. for their measurements. The correct error to report for a mean value is the standard error on the mean, not the standard deviation, as the former gives a more meaningful measure of the uncertainty in the average value.

6. The length values expected for the unfolding of each domain do not seem to be listed in the manuscript, making it difficult to judge whether the values measured were in fact reasonable. Please include these values in a table, and describe how they were obtained.

7. Page 4, lines 137-140: Please avoid combining results and interpretation at the same time—interpretation should be provided in the Discussion section. I think what the authors are saying is that when pulling the FimH domain, they see two unfolding transitions, which they then attribute to two different sub-domains. What evidence is there that this interpretation is correct? One can't simply assert this interpretation without providing the evidence that backs it up.

8. The conclusion about the hierarchy of mechanical strength in the pilus subunits seems reasonable, but it's possible that the interfaces between the subunits might change this picture. Measuring a construct containing all subunits assembled into a complete fibrillum (as suggested in the comment above) would help make this conclusion more robust.

9. Page 4, line 157: I'm not sure what “measuring FimH in its reduced state was elusive” is supposed to mean—that the reduced state was hard to create? That the reduced state was hard to observe once created? Please clarify.

10. Page 5, lines 190-194: While I am sympathetic to the claim that the mechanical folding assay is biologically relevant because the protein is unfolded as it is secreted through the membrane pore, it is not as well founded as the author may believe. The problem is that in the AFM refolding assay, the protein chain starts off completely unfolded all along its length, and all parts of the chain can begin the refolding at the same time. In contrast, when the protein translocates through the pore, it can begin to refold from the end as soon as that end has passed through the pore, in a manner analogous to what happens during co-translational folding (when the N-terminal parts of the protein can fold before the C-terminal parts have left the exit pore of the ribosome). The mechanical assay thus mimics the biological situation, but does not reproduce it perfectly. A more nuanced statement would be appropriate.

11. Page 7, line 284: In a similar vein, it is incorrect to state that the measurements of FimG with both FimC and DsbA “reproduce the actual conditions that pilus domains encounter in the periplasm.” The FimG domain is normally part of a quaternary complex, it is not attached to IG27 domains and an AFM cantilever, and many other types of molecules beyond FimC and DsbA are present. A more nuanced statement is needed.

12. In the refolding assay, one concern could be that the Ig27 domains influence the refolding. Please discuss this issue and how it may affect the interpretation.

13. The analysis of effects of FimC on the refolding kinetics of reduced FimG is not convincing, because of the wide spread of the results without FimC present. As far as I can see, the rate difference comes entirely from the first data point, but the variation from point to point at long times (when the amount of refolding is supposed to be constant) is larger than the difference between the final amount of refolding and the amount of refolding at the first time point. As a result, one can't have confidence in the fit to the data. The authors should measure sufficient additional time points that the exponential rise of the curve is clearly defined by the data. The other curves in Suppl Fig 8 are more convincing. Note that this curve with suspect datapoints is the only one that appears to have a different rate (the values of the others are all the same within error), hence it's important to clarify if the difference is real or an artifact of insufficient data.

14. Page 6 line 230: If two processes have different rates, it's impossible for the probability of refolding to be identical. I presume the authors mean that the refolding probabilities are the same once the processes reach equilibrium.

15. Page 6, paragraph 2: It's unclear what is being claimed here--is the issue that part of FimG is unfolding at 160 pN? Then why doesn't an intermediate show up in the extension records? Please clarify. On a related note, the design of the force-jump assay (going first to 160 pN before 300 pN) seems a little odd. Why not jump straight to 300 pN?

16. In the same paragraph, the authors claim that the 2-s pulse at 160 pN alters the rates registered at 300 pN. This statement seems to imply that the unfolding is not a Markovian process (i.e. there is some kind of memory effect involved), in contrast to the general view of protein folding/unfolding. What is the evidence for this claim?

17. Page 6 line 242: The authors state that the force-jump results “correlate” with the force-ramp results. Please show that the results obtained from these two complementary measurements are in fact quantitatively self-consistent. Analysis like that in Dudko et al PNAS 2008 shows how to map kinetics from force-ramps onto kinetics from force-jumps and constant-force data.

18. Discussion, line 310: The authors claim that their results demonstrate that “bacteria submitted to shear forces needs [sic] to compensate and distribute the mechanical stress along the pilus.” It is unclear where this idea comes from; what support for it is there in the results presented?

19. In Figs. 3c and 3e, it is odd to show time-varying data in a bar graph. Please replace the bar graphs with a proper graph plotting the observations as a function of time on the abscissa.

20. In Suppl Fig 3, please use the same units for the force as in the actual AFM data (pN, the SI unit of force, instead of kJ/mol/nm). If I have it right, the conversion should be roughly $2.5 \text{ kJ/mol/nm} = 4 \text{ pN nm}$.

Reviewer #2 (Remarks to the Author):

Applying single molecule force spectroscopy to donor-strand self-complemented Type I pilin subunits, the authors demonstrate that subunits FimA, FimF, and FimG are mechanically extremely strong (with FimA measuring ~ 500 pN), and that the tip subunit FimH (in particular the lectin domain) is less so. They propose this hierarchy helps maintain strong attachment up to the point at which detachment is preferred over breakage of the filament. The results are corroborated with molecular dynamics. The experiments are convincing and the model nicely parallels the catch-bond mechanism of adhesion. Separately it is shown that reduced subunits are mechanically weaker than oxidized ones.

Secondly, the authors suggest that DsbA serves not only an oxidoreductase but also a chaperone role. This is a conclusion I am not convinced of (see below).

The overall study is linked to the biology of the pilus and provides new insight into the mechanical stability of Type I pili and into the respective roles of periplasmic folding proteins DsbA and FimC.

Major comments:

1. The experimentally measured rupture forces are attributed to "the strength of the beta strand interaction" (line 129-130). But it is not clear how this conclusion is drawn?
2. The "worm like chain" appears only in the figure legend and needs to be described at least cursorily in the text. Why, for example, are there two WLC curves for each subunit (one folded, one unfolded)?
3. In Figure 2 it would be helpful to show on at least one plot where ΔL_c is measured (keeping in mind that not all readers of Nature Communications will be familiar with this technique).
4. Supplementary Figure 3 seems central to the results and could be moved to main paper.
5. Double check the axes for Supplementary Figure 4 – is x-axis time or extension?
6. Although the Discussion explains that it's the trend and not the absolute numbers that must be compared between experiment and simulation, is it not possible to at least use the same units for the graphs (pN vs. kJ/mol/nm)?
7. Supplementary Figure 6 is an important demonstration of the reversible folding of FimG. I would put it in the main manuscript.
8. The "Monitoring oxidation of individual FimG domains by DsbA" section on p. 6 is very wordy and should be shortened to make the point more clearly. The point seems to be DsbA recognizes and serves as an oxidoreductase for FimG. The authors claim that DsbA serves independently as a FimG chaperone. I don't understand how the data support this second claim. Wouldn't FimG folding probability be higher with an oxidoreductase than without if the disulfide bond formation steers the folding pathway? Perhaps this hypothesis could be more directly tested by measuring refolding of FimG(S-S) in the presence and absence of DsbA, or by comparing folding probabilities in the presence of a catalytically inactive DsbA vs. no DsbA?
9. The custom written Igor Pro code should be made available.
10. Clarify (and reference) that the idea of self-complemented subunits has been described numerous times previously.

Response to Reviewer #1

First, we want to thank the reviewer for his/her insightful comments. We have taken them very seriously and have done a significant amount of work to address them, including new experimental data and analysis. We acknowledge that our manuscript is now considerably better. We are also happy to see that this reviewer considers our work interesting, relevant, finds our experimental design clever, our measurements well done and sufficiently relevant for *Nature Communications*. He/she finds some flaws that we have addressed in the following lines. In blue, the literal comments from the reviewer:

“The most important concern regards the design of the study presented here. The authors have previously studied the unfolding of the FimG domain. The current manuscript extends that initial work in two directions: (i) by comparing the unfolding of FimG to that of FimH, FimF, and Fim A; and (ii) by looking at the refolding of FimG under different conditions. In my view, these two components each represent interesting but incremental advances over the previous work, in two different directions. Adding them together does not amount to the “complete mechanical characterization of the pilus” that the authors promise in the abstract. For a complete characterization, one would expect to investigate not only the unfolding of each of the domains separately, but also how each domain behaves in the context of a complete pilus, in order to understand the role of the full interfaces between the domains (not simply the effects of β -complementation, which are captured here).”

As the reviewer points out we have previously presented results for FimG, where we observed a very high mechanical stability for this protein. However, from the conclusions in that initial work we could not predict one of the most exciting contributions from the current work, i.e. the hierarchical mechanical organization of the pilus. For this reason, we cannot agree with the referee that the contribution of the current work is incremental. This is a completely unexpected and surprising result that has not even been predicted before. Here, we wanted to describe the origin of the stability and pilus design by studying the folding of the domains and the establishment of the interactions that confer such high mechanical stability with the intervention of other proteins such as DsbA and FimC. These two proteins are directly related to the two main components of the mechanical stability of the domains: the disulfide bonds and the β -strand complementation. In this sense, this work is one of the very few papers where the mechanical stability of a relevant protein such as the pilus is approached in its full dimension, that is, origin and features. For us, this is a complete description of pilus subunits folding, maturation and attachment. Also, we would like to stress that the folding of Fim domains has not been studied before in any form, with or without DsbA and FimC.

We see the point of the reviewer regarding the suggestion of studying more pilus domains to observe interfaces between domains and we have done so in this review process, but we want to stress that we have concentrated in the β -strand for two reasons. First, there is very little information about it despite being the most critical one. Second, other interactions between domains have been already reported. In this sense, the uncoiling of the FimA bundle as well as the formation of catch-bonds has been extensively studied by scientists Wendy Thomas, Evgeny Sokurenko and Viola Vogel (1-6). We have included this now in the text.

However, the comment from the reviewer brought up a very interesting question. Are there additional, quaternary interactions between the tip domains FimF-FimG-FimH? We further verified such possibility by constructing a new polyprotein sporting the full tip fibrillum array, (I91)₂-FimF-FimG-FimH-(I91)₂. We could only work with this protein by changing the operational mode in our AFM to force-ramp that allows a controlled force scanning to determine the mechanical stability of the array. From the experiments reported in the new Figure 3, we concluded that additional steps are observed suggesting a quaternary interaction between domains but considerably weaker than the β -strand complementation. This interaction seems to be similar to that found in the uncoiling of the FimA bundle. These new experiments also allowed a better resolution of the hierarchical mechanical stability of the pilus.

Overall, we tackle two of the relevant interactions of the pilus and we do it by linking them through the maturation process of β -strand and disulfide bond formation occurring within the

folding process. Amongst them we highlight the importance of the β -strand complementation because it is the only non-reversible interaction. Once it is broken, the pilus integrity is lost.

One would also expect to see a study of the refolding of more than just the FimG domain, but also the other domains. Indeed, the refolding of the FimH domain would seem to be the most important to understand, since it is the least mechanically stable domain and hence the one that should unfold and refold most frequently when the pilus is under tension. A truly complete mechanical characterization containing such additional measurements would, in my view, be suitable for publication in Nature Communications.

We totally agree with the reviewer that domain FimH is very important when considering the mechanics of the pilus and its attachment and we have included new experiments with it. From the folding point of view, only one domain can actually fold and refold when the pilus is under tension, and this is the lectin subdomain of FimH (FimH_L), which is located at the very tip and is responsible for the catch-bond attachment (see Figure 1a in main text). It would be interesting to see how fast the refolding process of FimH_L is, given that it is a crucial link in the pilus attachment and the unfolding may happen under extreme mechanical loads dampening the adhesion ability of the pilus. To investigate such process, we have performed new single-molecule force-clamp refolding experiments with FimH paying special attention to the more external FimH_L. A hypothetical situation of the pilus under tension would occur outside of the bacterium, where FimC and DsbA are not present. To reproduce that situation we have studied the (un)folding of FimH with both subdomains oxidized as they are found in a natural functional pilus. We designed force-clamp experiments with two force pulses. The first one at 60 pN triggers unfolding of the weaker FimH_L. The second one at 250 pN triggers unfolding of the pilin subdomain of FimH (FimH_P). We then quench the force to zero for 10 s. We found that the subdomain FimH_L shows higher probability of refolding. This fast refolding makes sense in a hypothetical situation in which this domain is unfolded and has to recover quickly. We have now reported this in the text and in new figure 6.

It is important to consider that FimH_P is β -strand complemented and folds in the periplasm with the intervention of DsbA and FimC. Once the pilus is assembled, FimH_P will not unfold again because that would trigger the rupture of the β -strand which means the loss of the whole pilus chain. This is also true for FimA, FimF and FimG. Our folding studies are centered in the role of DsbA and FimC on the formation of the mechanically relevant elements, i.e., disulfide bonds and shared β -strands. We do not expect DsbA and FimC to have a different mechanism for different Fim domains in the process of oxidation and donor-strand exchange.

Other issues to be addressed:

• line 51: instead of "full tertiary structure" I think the authors mean "full quaternary structure," given that several different domains are involved.

We have changed the statement for "full structural stabilization" given that quaternary stabilization may also refer to an additional interaction besides the β -strand.

• line 77: "nanomechanical architecture" is an overly vague phrase that is not meaningful. Strictly, the 'mechanical architecture' is just the structure, hence 'nanomechanical architecture' really just means the atomic-scale structure, which has indeed been solved for the pilus (it's shown in Fig 1A).

We see the point of the reviewer, but it is recognized that the word mechanical refers to the structural contacts that provide mechanical stability to the protein. In general, only a portion of the structure is truly responsible for such mechanical stability. This portion is generally referred to as the mechanical clamp (7). When we say the "mechanical architecture" we mean the structure of the elements that are directly responsible for the mechanical stability. This expression has been used before in a similar context (8). Puchner and Gaub defined mechanical architecture as "the way in which mechanical elements are arranged and defined" (9). In the case of the pilus assembly this structural elements are related to the different β -strands that extend from one domain to another to complement folding and provide the mechanical stability. The β -strands are thus part of the mechanical clamp, which we also

corroborate in our molecular dynamics simulations. The suffix “nano” only means that this structure is in the nanoscale.

• *line 79: “the mechanical resistance of the pilus” would be better expressed as “the resistance of the pilus to mechanical loads”*

We acknowledge change of the sentence in the text.

• *line 81: “the mechanical resistance of the subunits decreases as they approach to the tip” implies that the ‘mechanical resistance’ changes as the subunits move in space closer to the tip, which I do not believe is what the authors mean to say. Instead, the ‘mechanical resistance’ of the domains located close to the tip is lower than that of the domains located farther away*

The reviewer is absolutely correct. We have changed the paragraph.

• *Multiple locations: “therein” is used in situations where it is inappropriate.*

We have changed this.

2. *On a related note:*

• *The phrase “a mechanical hierarchy decreasing from the pilus rod to the tip” in the abstract is unclear.*

We have changed this.

• *Please spell out acronyms in the main text, not in the abstract.*

We have changed this.

• *Page 4, line 153: “non-disulfided” is not a word.*

We have corrected the word

3. *The first paragraph of the introduction jumps straight into the details of the structure of the pilus. It would be more effective to start by describing the problem that is being addressed in this work, before describing the background information about the pilus. Page 2, line 43: It would be helpful to reference Fig 1A (structure of pilus) here.*

Following this advice, we have rewritten the introduction and now the problem is first presented. There are many elements of the introduction that have been changed to provide a better view of the importance of our work.

4. *The authors use the term “atomic force spectroscopy” instead of the term that is almost universally used in the literature, atomic force microscope. There is no value to be gained by renaming a common tool with some bespoke label, only confusion. Please use the standard terminology instead.*

We have changed to AFM. There is now a tendency to name AFS to spectrometers without the possibility of being a real imaging microscope. But it is true that AFM is more common.

5. *The authors report mean \pm S.D. for their measurements. The correct error to report for a mean value is the standard error on the mean, not the standard deviation, as the former gives a more meaningful measure of the uncertainty in the average value.*

Although in AFM studies it is common to report standard deviation, the reviewer is correct that a measure of the standard error of the mean may be more meaningful. We have updated all our force data to report this parameter. However, we have maintained SD for contour length measurements given that SEM would report values below the resolution of the AFM.

6. *The length values expected for the unfolding of each domain do not seem to be listed in the manuscript, making it difficult to judge whether the values measured were in fact reasonable. Please include these values in a table, and describe how they were obtained.*

We have now included the expected values for contour length for each domain in the Supplementary Table 1. In Force-extension mode, the calculation is made by multiplying the number of residues by 0.4 nm and subtracting the length of the folded domains which is 4-5 nm, following reference (10).

7. Page 4, lines 137-140: Please avoid combining results and interpretation at the same time—interpretation should be provided in the Discussion section. I think what the authors are saying is that when pulling the FimH domain, they see two unfolding transitions, which they then attribute to two different sub-domains. What evidence is there that this interpretation is correct? One can't simply assert this interpretation without providing the evidence that backs it up.

In the crystal structure, FimH is clearly composed by two subdomains (see Figure 1a in main text and PDB: 4J3O), the lectin subdomain (FimH_L) and the pilin subdomain (FimH_P). As expected, we observe them both as two clearly differentiated peaks with contour length increments that agree with their size. The lectin subdomain is where the binding to D-mannose occurs, whereas the pilin subdomain is β -strand complemented. The difference in mechanical stability is the expected results given that only the pilin subdomain is β -strand complemented. We have clarified this in the text.

8. The conclusion about the hierarchy of mechanical strength in the pilus subunits seems reasonable, but it's possible that the interfaces between the subunits might change this picture. Measuring a construct containing all subunits assembled into a complete fibrillum (as suggested in the comment above) would help make this conclusion more robust.

We want to thank the reviewer for this interesting observation as we prove through new experiments that the domains seem to establish an additional interaction other than the β -strand complementation. Following the reviewer advice, we have made a polyprotein with the domains that compose the fibrillum, which are FimF-fimG-FimH (Figure 1a). The polyprotein used is (I91)₂-FimF-FimG-FimH-(I91)₂. Working with this polyprotein is especially difficult given that the high mechanical stability of the three Fim domains competes with the detachment of the protein. To prevent that, we decided to work in the force-ramp mode that allows linear scanning of the force, exposing the protein slowly to a progressive force, which triggers first the unfolding of weaker domains. The results are provided in a new figure in the main text (Figure 3) showing that apart from the β -strand connection between domains, a new weaker interaction emerges similar to that found in the uncoiling of the FimA bundle (11). Inherently, these experiments also allowed better resolution in force measurements for better definition of the mechanical hierarchy of the pilus. In FX experiments FimF and FimG are quite similar; however, force-ramp shows that FimF unfolds at slightly higher force than that of FimG. Thus the hierarchy is defined as FimA>FimF>FimG>FimH_P>FimH_L, which is the exact sequence of the pilus. Nevertheless, these experiments demonstrate that the β -strand interconnection still dominates the overall mechanical stability of the tip fibrillum. Again, we appreciate the comment from the reviewer that helped us to discover novel aspects in the complex mechanical design of the pilus.

9. Page 4, line 157: I'm not sure what "measuring FimH in its reduced state was elusive" is supposed to mean—that the reduced state was hard to create? That the reduced state was hard to observe once created? Please clarify.

We mean that it was difficult to observe the unfolding of FimH in the reduced state. We think that without the disulfide bond, the structure of FimH is less stable and in most cases might not even have a proper folding. Given that this is a speculation, we have removed it from the text.

10. Page 5, lines 190-194: While I am sympathetic to the claim that the mechanical folding assay is biologically relevant because the protein is unfolded as it is secreted through the membrane pore, it is not as well founded as the author may believe. The problem is that in the AFM refolding assay, the protein chain starts off completely unfolded all along its length, and all parts of the chain can begin the refolding at the same time. In contrast, when the protein translocates through the pore, it can begin to refold from the end as soon as that end has passed through the pore, in a manner analogous to what happens during co-translational folding (when the N-terminal parts of the protein can fold before the C-terminal parts have left the exit pore of the ribosome). The mechanical assay

thus mimics the biological situation, but does not reproduce it perfectly. A more nuanced statement would be appropriate.

We acknowledge that the reviewer is absolutely correct in this observation. What we mean is that the AFM experiments are in our view the closest rendition to the real biological situation of a Fim domain unfolding and refolding, traveling extended through the pore to the periplasm, more so than any bulk experiment in a test tube. We have now written this more carefully in the text.

11. Page 7, line 284: In a similar vein, it is incorrect to state that the measurements of FimG with both FimC and DsbA “reproduce the actual conditions that pilus domains encounter in the periplasm.” The FimG domain is normally part of a quaternary complex, it is not attached to IG27 domains and an AFM cantilever, and many other types of molecules beyond FimC and DsbA are present. A more nuanced statement is needed.

The reviewer is also correct here and we have attenuated our statement. What we mean again is that our experimental design is intended to involve the important elements that are known to be present in the oxidative folding of Fim domains. Of course, the cellular environment is crowded and no AFM tips are present, but we hope the reviewer will agree with us in that trying to intervene in a system implies technical limitations that for the most part are unavoidable.

12. In the refolding assay, one concern could be that the Ig27 domains influence the refolding. Please discuss this issue and how it may affect the interpretation.

We agree with the reviewer that tethering of I91 molecules and Fim domains could potentially have an effect on the folding of the monomers. However, this is a debate that was experimentally refuted in a work by Garcia-Manyes et al in 2007, by probing that the refolding kinetics of a single I27 (now named I91) monomer is equivalent to that of the polyprotein made of the same domain (12). If the I91 domain does not interfere with itself, is not expected that it interferes with a completely different molecule such as FimG.

13. The analysis of effects of FimC on the refolding kinetics of reduced FimG is not convincing, because of the wide spread of the results without FimC present. As far as I can see, the rate difference comes entirely from the first data point, but the variation from point to point at long times (when the amount of refolding is supposed to be constant) is larger than the difference between the final amount of refolding and the amount of refolding at the first time point. As a result, one can't have confidence in the fit to the data. The authors should measure sufficient additional time points that the exponential rise of the curve is clearly defined by the data. The other curves in Suppl Fig 8 are more convincing. Note that this curve with suspect datapoints is the only one that appears to have a different rate (the values of the others are all the same within error), hence it's important to clarify if the difference is real or an artifact of insufficient data.

The reviewer is absolutely correct that the scattering in the data may appear an artifact, perhaps due to insufficient data, especially at 10 s. We have included more experimental data at all times but with special attention at 10 s, and have reanalyzed all data sets. We have found that the actual rate of refolding of reduced FimG in the absence and presence of FimC are quite similar, indicating that FimC has no effect on reduced FimG and only acts in oxidized substrates. The new figure 4 has been updated.

14. Page 6 line 230: If two processes have different rates, it's impossible for the probability of refolding to be identical. I presume the authors mean that the refolding probabilities are the same once the processes reach equilibrium.

The reviewer is absolutely correct. What we mean is that at longer times, once equilibrium is reached, the probability of refolding of reduced FimG in the presence and absence of FimC is similar to that found for reduced FimG with no FimC but smaller than FimG with FimC, which suggested that FimC actually recognizes disulfide bonded domains. We have clarified this in the text.

15. Page 6, paragraph 2: It's unclear what is being claimed here--is the issue that part of FimG is unfolding at 160 pN? Then why doesn't an intermediate show up in the extension records? Please

clarify. On a related note, the design of the force-jump assay (going first to 160 pN before 300 pN) seems a little odd. Why not jump straight to 300 pN?

We are happy to explain this. We apply 160 pN to trigger unfolding of I91. The design of the force-clamp experiments is such that allows the detection of all possible steps. If we jump straight to 300 pN the unfolding of I91 domains will occur so fast that will be technically unfeasible to discern the four individual steps that serve as fingerprint. At 160 pN we are able to see each I91 unfolding event. We have successfully used this protocol before for enzymatic and folding studies in proteins (13-15).

16. In the same paragraph, the authors claim that the 2-s pulse at 160 pN alters the rates registered at 300 pN. This statement seems to imply that the unfolding is not a Markovian process (i.e. there is some kind of memory effect involved), in contrast to the general view of protein folding/unfolding. What is the evidence for this claim?

The reviewer is absolutely correct. We acknowledge that there is no evidence that implies that the unfolding is not Markovian. We have removed that sentence and used single exponential fits.

17. Page 6 line 242: The authors state that the force-jump results “correlate” with the force-ramp results. Please show that the results obtained from these two complementary measurements are in fact quantitatively self-consistent. Analysis like that in Dudko et al PNAS 2008 shows how to map kinetics from force-ramps onto kinetics from force-jumps and constant-force data.

This is another excellent point from the referee. Indeed, to make the agreement between force-extension and force-clamp results in smFS one can use the Dudko-Hummer-Szabo method that results in a direct mapping between the distribution of rupture forces $P(F)$ and the force dependence of the lifetime, $\tau(F)$ (16). We have carried out this analysis using the force extension data at 400 nm/s for FimG, both for the reduced and oxidized variants. The results are included in the figure 1 below. The top panel shows the normalized histograms for unfolding forces, $P(F)$, for the oxidized (dark green) and reduced (light green) forms of FimG. Using expression [2] from Dudko et al, one can calculate $\tau(F)$ from these histograms and the loading rate, \dot{F} . Estimates for $\tau(F)$ for oxidized and reduced forms are shown in the bottom panel of the figure (triangles). We compare them directly with the value of the lifetime measured in our force clamp experiments at 300 pN (circle). As can be seen, the agreement between force clamp and force extension for FimG is excellent. For simplicity we use the Bell expression that sensibly captures the trends observed in our datasets. We note that for the calculation of the loading rate we did not incorporate explicitly the effects of the linkers, as they have been shown to be negligible in the high force regime.

Despite the agreement between force extension and force clamp, we have decided not to include these results in our manuscript on the basis of scientific rigor. First, the formalism used here is particularly useful when histograms of rupture forces are available at a range of pulling speeds greater than one order of magnitude, and constant force measurements are performed at multiple forces. If this were the case, one would see that all the data collapses into a single, master curve in the plot of the force dependent lifetimes. Unfortunately, it would be prohibitive to make those measurements for the systems under study, which is also out of the scope of this study. Second, to derive intrinsic microscopic parameters of the proteins from the force dependent lifetimes (e.g. the intrinsic rate and the distance to the transition state in Bell's formalism and even the intrinsic free energy barrier in Dudko's own method (17)), would imply very long propagations of much more than 100 pN from the available range of forces, making our extrapolated estimates unreliable.

We expect the referee will agree that given these considerations it is preferable not to report the analysis as part of the paper, which has in any case a more biological (as opposed to biophysical) focus. This may be, however, an interesting aspect to study in future work from our laboratory.

Figure 1. Mapping between distribution of rupture forces $P(F)$ and force dependence of the lifetime, $\tau(F)$. Force extension experiments are shown as triangles and force-clamp experiment as a circle.

18. Discussion, line 310: The authors claim that their results demonstrate that “bacteria submitted to shear forces needs [sic] to compensate and distribute the mechanical stress along the pilus.” It is unclear where this idea comes from; what support for it is there in the results presented?

Given the coil structure, the whole pilus acts as a shock absorber, and as such the force is distributed along the pilus. The reviewer is correct; this idea does not come from our experiments. However, our results show that the non-reversible interaction, i.e., the β -strand interconnection, is much higher than the catch-bond and the uncoiling of the FimA rod. Having this very high mechanical connection between domains certainly allows for the pilus to behave as a shock absorber. To see this let’s imagine that the β -strand interconnection were instead in the order of 100-150 pN. Then, the β -strand disconnection would happen at similar forces than the uncoiling of the rod. In such situation there will be no shock absorber function because the pilus would simply break. This is what we mean when we say that our experiments support the idea of a shock absorber that distributes force. We have now clarified this in the text

19. In Figs. 3c and 3e, it is odd to show time-varying data in a bar graph. Please replace the bar graphs with a proper graph plotting the observations as a function of time on the abscissa.

We have replaced both plots to reflect probability of refolding versus time.

20. In Suppl Fig 3, please use the same units for the force as in the actual AFM data (pN, the SI unit of force, instead of kJ/mol/nm). If I have it right, the conversion should be roughly $2.5 \text{ kJ/mol/nm} = 4 \text{ pN nm}$.

We have followed the reviewer advice and have unified the units used.

Response to Reviewer #2.

We are happy to see that this reviewer finds that our experiment “are convincing” and our work “provides new insights into the mechanical stability of the Type 1 pili and into the respective roles of periplasmic folding proteins DsbA and FimC”. We sympathize with the reviewer and find his/her comments helpful for our work. In what follows, we answer his/her concerns:

1. The experimentally measured rupture forces are attributed to “the strength of the beta strand interaction” (line 129-130). But it is not clear how this conclusion is drawn?

Our constructs following the self-complementation strategy are designed in a way that the force vector has the direction of the added β -strand. This is shown in Figure 1c. We have updated this figure to be clearer, adding the vector force with the arrows indicating the pulling direction along the β -strand. When we say the strength of the β -strand, we actually mean the force needed for the rupture of such β -strand. To make it clear we have rewritten the sentence as “we attribute this force to the rupture of the β -strand interaction”. This is clear by the contour length measured of 40-45 nm for oxidized domains which corresponded to the unraveling of the domain starting by the rupture of the β -strand. We calculate this by multiplying the number of residues by 0.4 nm/residue, and subtracting about 4-5 nm of the folded domain, following reference (10). This gives a total length which is in close agreement with the measured length. This demonstrates that the rupture of the β -strand establishes the unraveling starting point. We have now clarified this in the text.

2. The “worm like chain” appears only in the figure legend and needs to be described at least cursorily in the text. Why, for example, are there two WLC curves for each subunit (one folded, one unfolded)?

This is a valid point. We have now updated the methods section and text to include the usage of the WLC and a proper reference. The two lines are drawn between a totally unfolded domain to estimate the contour length increment upon unfolding. Therefore the two lines delimitate the fully extended domain. This is the general way used to estimate the length of each domain upon unfolding.

3. In Figure 2 it would be helpful to show on at least one plot where ΔL_c is measured (keeping in mind that not all readers of Nature Communications will be familiar with this technique).

We appreciate this point and agree with the reviewer. We have updated figure 2 to include an indication of ΔL_c in the first experimental trace to make clear where that parameter is coming from.

4. Supplementary Figure 3 seems central to the results and could be moved to main paper.

We respectfully see it differently, as this figure basically provides a general agreement with the experimental data which in our opinion are the central results. All the molecular dynamics simulations in the present work provide corroborative results to our experiments; this is why we decided to show experiments in the main text and the simulations in supplementary information.

5. Double check the axes for Supplementary Figure 4 – is x-axis time or extension?

The trace represents force versus extension. We thank the reviewer for this comment as we have detected a mistake in the legend mentioning time instead of extension; our apologies. The mistake is now fixed.

6. Although the Discussion explains that it's the trend and not the absolute numbers that must be compared between experiment and simulation, is it not possible to at least use the same units for the graphs (pN vs. kJ/mol/nm)?

Yes, it is possible and we have done so. The figures have been updated with consistent units.

7. Supplementary Figure 6 is an important demonstration of the reversible folding of FimG. I would put it in the main manuscript.

In fact, Supplementary Figure 6 (now Supplementary Figure 4) does not really show reversible unfolding. The reversible unfolding is actually shown in figure 4 and 5 where the unfolding of FimG is monitored in first and second force pulses. In Supplementary Figure 4 we only observe unfolding in the first pulse but not in the second, suggesting that the protein did not refold during the quenching time.

8. The “Monitoring oxidation of individual FimG domains by DsbA” section on p. 6 is very wordy and should be shortened to make the point more clearly. The point seems to be DsbA recognizes and serves as an oxidoreductase for FimG. The authors claim that DsbA serves independently as a FimG chaperone. I don’t understand how the data support this second claim. Wouldn’t FimG folding probability be higher with an oxidoreductase than without if the disulfide bond formation steers the folding pathway? Perhaps this hypothesis could be more directly tested by measuring refolding of FimG(S-S) in the presence and absence of DsbA, or by comparing folding probabilities in the presence of a catalytically inactive DsbA vs. no DsbA?

This is a very valid point from the reviewer and we have included new data and analysis to address it. To further test that DsbA can serve as a chaperone even if not involving disulfide bond formation, we have collected new data in which we monitor folding of FimG in the presence of oxidized DsbA but with no formation of disulfide bond, that is, there is no oxidase activity by DsbA (see new Figure 5 and supplementary figure 7). Even in this situation, we have observed a folding probability of about 50%. This contrasts with our experiments in the absence of DsbA in which the refolding probability of FimG is only 20% and most traces look like the one in Supplementary Figure 4. This new analysis probes the idea of the reviewer. However, from the data in Figure 5, it is true that the formation of the disulfide bond increases even more the folding probability of FimG without the need of FimC. Overall, we believe that DsbA performs a chaperone activity mainly throughout its oxidase activity, but can do it even with no oxidase activity. This is a novel observation because it was believed that DsbA has little to do with the folding and only renders an oxidized unstructured substrate for FimC to complete folding. FimC has been generally viewed as the real chaperone and DsbA as an oxidant (18). We believe that DsbA actually renders an oxidized substrate with a quasi-folded structure for FimC to bind and act as a transporter.

9. The custom written Igor Pro code should be made available.

We have included a statement in the Material and Methods section in which we make available this software to the scientific community upon request.

10. Clarify (and reference) that the idea of self-complemented subunits has been described numerous times previously.

The idea of self-complementation has been described before for FimH, we have now included this reference in our list (19). However, as far as we know, there is not much information in the literature following that strategy. What is totally new in our work is that we use the idea to study the mechanical stability of the pilus domains.

References

1. M. Forero, O. Yakovenko, E. V. Sokurenko, W. E. Thomas, V. Vogel, Uncoiling mechanics of Escherichia coli type I fimbriae are optimized for catch bonds. *PLoS Biol* **4**, e298 (Sep, 2006).
2. W. E. Thomas, E. Trintchina, M. Forero, V. Vogel, E. V. Sokurenko, Bacterial adhesion to target cells enhanced by shear force. *Cell* **109**, 913 (Jun 28, 2002).
3. E. V. Sokurenko, V. Vogel, W. E. Thomas, Catch-bond mechanism of force-enhanced adhesion: counterintuitive, elusive, but ... widespread? *Cell Host Microbe* **4**, 314 (Oct 16, 2008).
4. L. M. Nilsson, W. E. Thomas, E. Trintchina, V. Vogel, E. V. Sokurenko, Catch bond-mediated adhesion without a shear threshold: trimannose versus monomannose interactions with the FimH adhesin of Escherichia coli. *J Biol Chem* **281**, 16656 (Jun 16, 2006).
5. P. Aprikian *et al.*, Interdomain interaction in the FimH adhesin of Escherichia coli regulates the affinity to mannose. *J Biol Chem* **282**, 23437 (Aug 10, 2007).
6. I. Le Trong *et al.*, Structural basis for mechanical force regulation of the adhesin FimH via finger trap-like beta sheet twisting. *Cell* **141**, 645 (May 14, 2010).
7. M. Sikora, M. Cieplak, Mechanical stability of multidomain proteins and novel mechanical clamps. *Proteins* **79**, 1786 (Jun, 2011).
8. H. Li, J. M. Fernandez, Mechanical design of the first proximal Ig domain of human cardiac titin revealed by single molecule force spectroscopy. *J Mol Biol* **334**, 75 (Nov 14, 2003).
9. E. M. Puchner, H. E. Gaub, Single-molecule mechanoenzymatics. *Annu Rev Biophys* **41**, 497 (2012).
10. S. R. Ainaravapu *et al.*, Contour length and refolding rate of a small protein controlled by engineered disulfide bonds. *Biophys J* **92**, 225 (Jan 1, 2007).
11. A. Tsigotaki, J. De Geyter, N. Sostaric, A. Economou, S. Karamanou, Protein export through the bacterial Sec pathway. *Nat Rev Microbiol* **15**, 21 (Jan, 2017).
12. S. Garcia-Manyes, J. Brujic, C. L. Badilla, J. M. Fernandez, Force-clamp spectroscopy of single-protein monomers reveals the individual unfolding and folding pathways of I27 and ubiquitin. *Biophys J* **93**, 2436 (Oct 1, 2007).
13. T. B. Kahn, J. M. Fernandez, R. Perez-Jimenez, Monitoring Oxidative Folding of a Single Protein Catalyzed by the Disulfide Oxidoreductase DsbA. *J Biol Chem* **290**, 14518 (Jun 5, 2015).
14. A. P. Wiita *et al.*, Probing the chemistry of thioredoxin catalysis with force. *Nature* **450**, 124 (Nov 1, 2007).
15. R. Perez-Jimenez *et al.*, Diversity of chemical mechanisms in thioredoxin catalysis revealed by single-molecule force spectroscopy. *Nat Struct Mol Biol* **16**, 890 (Aug, 2009).
16. O. K. Dudko, G. Hummer, A. Szabo, Theory, analysis, and interpretation of single-molecule force spectroscopy experiments. *Proceedings of the National Academy of Sciences* **105**, 15755 (2008).
17. O. K. Dudko, G. Hummer, A. Szabo, Intrinsic Rates and Activation Free Energies from Single-Molecule Pulling Experiments. *Physical Review Letters* **96**, 108101 (03/15/, 2006).
18. M. D. Crespo *et al.*, Quality control of disulfide bond formation in pilus subunits by the chaperone FimC. *Nat Chem Biol* **8**, 707 (Aug, 2012).
19. M. M. Barnhart *et al.*, PapD-like chaperones provide the missing information for folding of pilin proteins. *Proc Natl Acad Sci U S A* **97**, 7709 (Jul 5, 2000).

REVIEWERS' COMMENTS:

Reviewer #1 (Remarks to the Author):

The authors have made substantial changes to the manuscript to address the concerns that were raised by the reviewers. The measurements on the new construct and the refolding measurements of FimH are a nice addition and definitely help strengthen the conclusions. The manuscript is much improved, and I believe it is suitable for publication with only minor revisions needed to address some remaining issues.

1. Some statements are more general or more emphatic than they should be. For example, page 2 line 60: while some catch bonds detach at 150-180 pN, not all of them do (and of course the breaking force depends on the conditions of the measurement), so the sentence should be re-written more carefully. Page 3 line 98: "perfectly complements" is too strong, since nothing is ever perfect!

2. On page 4 line 141, the authors might want to reference very recent work by the Gaub lab (Milles et al Science 2018) that demonstrates a 2 nN unbinding force.

3. Page 4 line 155-159: these sentences are a little confusing, since they suggest that about 1/3 of the curves have very much lower force than the average. Are the low-force curves not included in the averaging reported in the first sentence? If they are, how is it possible to have 1/3 of the data with such a different force?

4. Line 187-189: the "force ramp" measurements need to be explained a little better. I assume that they involve ramping up the force at a fixed rate while using a force clamp to maintain the desired value at any given instant, is that correct? Is that how they differ from the measurements shown earlier where the force is ramped up at a non-constant rate simply by moving the cantilever at a constant speed? Please clarify.

5. Line 195-96: strictly speaking, it's the loading rate (rate of change of force) that's the basis of comparison, not the pulling speed, since the loading rate is the quantity that influences the unfolding force.

6. Lines 200-215: regarding the postulated quaternary interactions that lead to the 6-nm steps, does this length correspond to what would be expected given the known geometry of the linked domains?

7. Lines 365-367: This sentence doesn't make sense because it contrasts FimH_L with FimH_L; it apparently unfolds at both 60 and 160 pN?? I assume there is a typo somewhere and the second instance of FimH_L should be FimH_P?

8. In Table S1, I notice that both FimA reduced and FimF reduced have lengths that are nominally "significantly" different from the expected lengths (in one case, the difference is 3 standard deviations, in the other it is 6 standard deviations). The disagreement for FimA is probably just a statistical fluctuation, but the disagreement for FimF is large enough to indicate that the measured length disagrees with the expectation. Some comment about the discrepancy would be appropriate (perhaps due to underestimated error, or to an incorrect estimate of the expected length?).

9. The grammar and language are much improved, but there are still a few problems to fix. For example, on page 2 line 59: the word "lockers" is not appropriate in this context. There are a few other examples scattered in the Discussion section, too.

10. One issue that I don't feel was properly resolved related to point 12 in the first review: that the I91 domains might affect the refolding. The authors claim that work done comparing the refolding kinetics of a single I91 domain to those of a I91 tandem repeat protein prove that this concern is not justified. However, the logic here is not sound: the fact that the presence of one I91 domain does not influence the refolding of another I91 does **not**, in fact, prove that the presence of a I91 domain may not influence the folding of a different domain (here, the Fim domains), since there is no reason that the I91 and Fim domains would necessarily interact in the same way that I91 and I91 interact (or as the case would appear to be, don't interact). A counter-example is work done on disordered proteins inserted into I91 polyproteins (like that by Bruno Samori and colleagues about 10 years ago), which found that a disordered protein like α -synuclein can form structures that support hundreds of pN of force when placed in a polyprotein (something that is obviously not consistent with the protein remaining disordered!). To be clear, I think all that is needed here is just an acknowledgement of the possibility.

Reviewer #2 (Remarks to the Author):

The authors have done a conscientious job in responding to all of the comments of both reviewers, included collecting additional data and re-evaluating some of their statements. In my opinion the revised version is much stronger. I can agree to disagree on some fine points, and think this paper is ready to be published in Nature Communications.

Response to reviewer #1

We are delighted to see that this reviewer is happy with our revision and recommends our paper for publication. We have to say that his/her comments were quite inspiring and truly helped to improve our work. Here our response to the minor issues remaining:

1. Some statements are more general or more emphatic than they should be. For example, page 2 line 60: while some catch bonds detach at 150-180 pN, not all of them do (and of course the breaking force depends on the conditions of the measurement), so the sentence should be re-written more carefully. Page 3 line 98: "perfectly complements" is too strong, since nothing is ever perfect!

It is true. We have now changed the text. The detachment force of catch bonds statement is now in the discussion. As the reviewer specifies, we have mentioned that rupture force depends on the conditions used. We have also written the mentioned sentence in page 3 in a more cautious way.

2. On page 4 line 141, the authors might want to reference very recent work by the Gaub lab (Milles et al Science 2018) that demonstrates a 2 nN unbinding force.

We have included the reference.

3. Page 4 line 155-159: these sentences are a little confusing, since they suggest that about 1/3 of the curves have very much lower force than the average. Are the low-force curves not included in the averaging reported in the first sentence? If they are, how is it possible to have 1/3 of the data with such a different force?

The average force is reported for each peak. Most traces show a single peak for FimH_L of about 40 nm of length and 130 pN of unfolding force. This set is reported individually in Supplementary figure 1 as brown symbols. In addition, we often observe this domain with an intermediate of lower force. In these traces we observe two populations that are treated individually, the smaller peak of 6 nm and 98 pN, precedes a big one of 36 nm and 107. This is reported as two distinct groups of grey dots in Supplementary figure 1. We have clarified this in the text and supplementary figure 1.

4. Line 187-189: the "force ramp" measurements need to be explained a little better. I assume that they involve ramping up the force at a fixed rate while using a force clamp to maintain the desired value at any given instant, is that correct? Is that how they differ from the measurements shown earlier where the force is ramped up at a non-constant rate simply by moving the cantilever at a constant speed? Please clarify.

That is correct, in the force-ramp mode the force is ramped up in a totally controlled way by the feedback loop. This is the main difference with respect to force extension. We have now clarified this in the text.

5. Line 195-96: strictly speaking, it's the loading rate (rate of change of force) that's the basis of comparison, not the pulling speed, since the loading rate is the quantity that influences the unfolding force.

This is correct; we have changed pulling speed for loading rate.

6. Lines 200-215: regarding the postulated quaternary interactions that lead to the 6-nm steps, does this length correspond to what would be expected given the known geometry of the linked domains?

We believe so. The size of all Fim domains is quite similar and the 6 nm step is close to the value determined by Vogel and coworkers for FimA uncoiling, so we speculate that we are measuring a similar interaction.

7. Lines 365-367: This sentence doesn't make sense because it contrasts FimH_L with FimH_L; it apparently unfolds at both 60 and 160 pN?? I assume there is a typo somewhere and the second instance of FimH_L should be FimH_P?

That is correct, we have corrected the typo.

8. In Table S1, I notice that both FimA reduced and FimF reduced have lengths that are nominally "significantly" different from the expected lengths (in one case, the difference is 3 standard deviations, in the other it is 6 standard deviations). The disagreement for FimA is probably just a statistical fluctuation, but the disagreement for FimF is large enough to indicate that the measured length disagrees with the expectation. Some comment about the discrepancy would be appropriate (perhaps due to underestimated error, or to an incorrect estimate of the expected length?).

The values in Table S1 are within experimental resolution. The discrepancy between measured and theoretical ΔL_c comes from the fact that for theoretical calculations all residues are accounted. However, the actual mechanical clamp generally does not include all residues, as some of them, especially in the termini may not have a real contribution to the mechanical stability. For instance I91 has 89 residues, considering the length per residue, 0.4 nm and the length of the folded domain, around 4 nm, the expected ΔL_c would be 31.6 nm, but we always measure 28 nm, which is a well-known value. This is because there are some residues that do not have a real contribution to the stability, and are out of the mechanical clamp. The values that we report for some Fim domains may arise from a similar situation.

9. The grammar and language are much improved, but there are still a few problems to fix. For example, on page 2 line 59: the word "lockers" is not appropriate in this context. There are a few other examples scattered in the Discussion section, too.

We appreciate this comment. We have made the suggested change and others in the text.

*10. One issue that I don't feel was properly resolved related to point 12 in the first review: that the I91 domains might affect the refolding. The authors claim that work done comparing the refolding kinetics of a single I91 domain to those of a I91 tandem repeat protein prove that this concern is not justified. However, the logic here is not sound: the fact that the presence of one I91 domain does not influence the refolding of another I91 does *not*, in fact, prove that the presence of a I91 domain may not influence the folding of a different domain (here, the Fim domains), since there is no reason that the I91 and Fim domains would necessarily interact in the same way that I91 and I91 interact (or as the case would appear to be, don't interact). A counter-example is work done on disordered proteins inserted into I91 polyproteins (like that by Bruno Samori and colleagues about 10 years ago), which found that a disordered protein like α -synuclein can form structures that support hundreds of pN of force when placed in a polyprotein (something that is obviously not consistent with the protein remaining*

disordered!)). To be clear, I think all that is needed here is just an acknowledgement of the possibility.

As the reviewer points out, it is a good exercise to acknowledge such possibility, and we do, but it is also true that recent experiments have questioned the existence of such interactions. Perhaps, the most clear evidence was reported by Fernandez and coworkers that used magnetic tweezers to show that I91 domains in a polyprotein can fold individually one by one, with no apparent interaction between them (*Cell Rep* (2018), 4(6):1339). Additionally, experiments by Garcia-Manyes and coworkers have shown that I91 domains do not fold when they interact with Hsp40 and Hsp70, while these domains fold in the absence of such chaperones (*Sci Adv* (2016), 4(2)). All this comes to show that I91 domains fold on their own when they are in a polyprotein, and that an external interaction can indeed prevent folding. In our experiments, if the Fim domain were to interact with I91 we would likely see no refolding of either the Fim domain or I91 domains, but we see it. Again, we agree with the reviewer that the possibility exist, but we think that it may happen at a time scale that is out of our reach and that does not have a substantial effect on the folding of Fim and I91 domains.